# A small bacteriophage protein determines the hierarchy over co-residential jumbo phage in *Bacillus thuringiensis* serovar *israelensis*

Anja Pavlin[1,6], Anže Lovše[1,2,6], Gregor Bajc[1], Jan Otoničar[1], Amela Kujović[1], Živa Lengar[3], Ion Gutierrez-Aguirre[3], Rok Kostanjšek [1], Janez Konc[4], Nadine Fornelos[5] & Matej Butala [1✉]

*Bacillus thuringiensis* serovar *israelensis* is the most widely used biopesticide against insects, including vectors of animal and human diseases. Among several extrachromosomal elements, this endospore-forming entomopathogen harbors two bacteriophages: a linear DNA replicon named GIL01 that does not integrate into the chromosome during lysogeny and a circular-jumbo prophage known as pBtic235. Here, we show that GIL01 hinders the induction of cohabiting prophage pBtic235. The GIL01-encoded small protein, gp7, which interacts with the host LexA repressor, is a global transcription regulator and represses the induction of pBtic235 after DNA damage to presumably allow GIL01 to multiply first. In a complex with host LexA in stressed cells, gp7 down-regulates the expression of more than 250 host and pBtic235 genes, many of which are involved in the cellular functions of genome maintenance, cell-wall transport, and membrane and protein stability. We show that gp7 homologs that are found exclusively in bacteriophages act in a similar fashion to enhance LexA's binding to DNA, while likely also affecting host gene expression. Our results provide evidence that GIL01 influences both its host and its co-resident bacteriophage.

[1] Department of Biology, Biotechnical Faculty, University of Ljubljana, Ljubljana, Slovenia. [2] Genialis, Inc., Boston, MA, USA. [3] Department of Biotechnology and Systems Biology, National Institute of Biology, Ljubljana, Slovenia. [4] Theory Department, National Institute of Chemistry, Ljubljana, Slovenia. [5] Broad Institute of MIT and Harvard, Cambridge, MA, USA. [6]These authors contributed equally: Anja Pavlin, Anže Lovše. ✉email: matej.butala@bf.uni-lj.si

Most bacteria carry prophages. These are genomes of temperate bacteriophages that have critical roles in shaping bacterial ecology and in some cases enable the virulence of pathogens[1–3]. During infection, temperate phages can enter lysogeny (i.e., a dormant state within the bacterium) or activate the lytic cycle to produce new phage progeny. Temperate phages have evolved a range of mechanisms to promote these opposite lifestyles[4,5], although, how these are controlled and how the phage takes over the host processes is only understood for a small number of phages.

To achieve stable lysogeny, some temperate phages rely on their own transcription repressors with embedded autoproteolytic activity in the dimerization domain[6–8]. Upon DNA damage and SOS response induction, the active RecA filament (RecA*) that is formed on single-stranded DNA at sites of DNA damage triggers the self-cleavage and inactivation of the repressor; this drives the phage into the lytic cycle. In contrast, other phages encode for repressors where the DNA-binding activity is abolished by an anti-repressor protein[4,9,10] or a dedicated protease[5], both of which can be regulated through a variety of mechanisms. Other phages, such as tectivirus GIL01 that infects the spore-forming bacterium *Bacillus thuringiensis* serovar *israelensis*, a widely used biopesticide, do not encode their own repressor, but use the host LexA to establish the prophage quiescent state[11].

The GIL01 15-kb linear genome does not integrate into the host chromosome and has terminal proteins covalently linked to each 5'-end. GIL01 replicates its genome *via* a terminal protein-primed amplification using its own DNA polymerase[12]. The genome of GIL01 includes 30 open reading frames (ORFs) divided into two main modules that are transcribed in the same direction[13,14]. The tandem promoters *P1* and *P2* are located upstream of the eight genes (ORF1-ORF8) that constitute the module for proteins involved in the regulation of the GIL01 genome's replication and transcription. The promoter *P3* controls the expression of the downstream genes involved in the host recognition, the assembly of the capsid (which is composed of an outer protein layer and an inner protein-rich lipid membrane), and cell lysis[15]. In a previous study we established that the GIL01-encoded small proteins gp1 and gp7 are involved in lysogeny, and mutations in either of these proteins lead to the lysogenic state not being established[14,16]. Gp7 forms a complex with the host LexA repressor and acts as a scaffold to set the repressor in a conformation that is specific for binding to target DNA sites[16,17]. In doing so, gp7 acts as a co-repressor that subverts host LexA to occupy two operators at *P1*, one of which lacks a canonical LexA sequence motif. By expressing gp7, GIL01 uses LexA as a reliable indicator of the host's viability. In addition, gp7 has a dual action on RecA, by increasing LexA's affinity for the *recA* operator and by impairing the RecA-stimulated self-cleavage of LexA, which might alter the dynamics of the host SOS response[16,17].

Serovars of *B. thuringiensis* carry a diverse pool of self-replicating extrachromosomal DNA molecules. For instance, nine plasmids have been described for serovar *israelensis*, which include the well-characterized pBtoxis that encodes the entomopathogenic crystals produced during sporulation, and two prophages, i.e., the linear double-stranded DNA replicon GIL01 and the circular prophage pBtic235[11]. pBtic235 is a 235-kb low-copy-number element that is composed of plasmid and phage modules that are mainly encoded on the sense and anti-sense strands, respectively[11]. pBtic235 carries 287 genes and appears to influence the host bacterium's lifestyle, as it encodes predicted homologs of metabolite transporters, tellurium resistance, SOS-inducible phage functions, and stress-response and genome-maintenance functions, such as *uvrD* helicase. In addition,

pBtic235 encodes its own transcription factors, such as sigma-70 factor, several tRNA molecules, and determinants involved in genome replication, which combined with a large genome size, are a feature of jumbo phages[18]. This extrachromosomal element appears to transiently integrate its genome into the host chromosome, where it is believed to have a role in terminating the chromosome's replication[19]. The potential molecular cross-talk between pBtic235 and GIL01 and its effects on the lifestyle of *B. thuringiensis* serovar *israelensis* remain unstudied. Here, we describe how the GIL01-encoded co-repressor gp7 coordinates the induction of prophage pBtic235 and affects the overall transcription in *B. thuringiensis* serovar *israelensis*. On the computational side we provide a pipeline for a reproducible and transparent RNA-sequencing analysis.

## Results

**B. thuringiensis serovar israelensis and prophage pBtic235 change transcriptional programs in response to prophage GIL01 induction**. GIL01 establishes a lysogenic cycle in the infected *B. thuringiensis* serovar *israelensis* strain GBJ002, a 4Q7 derivative cured of all plasmids and three chromosomal prophages, except for the pBtic235 prophage (Supplementary Fig. 1)[19]. Our aim was to to study transcription in strain GBJ002 in presence of GIL01. To this end, we determined and compared the transcriptomes in the GIL01 and pBtic235 lysogenic strain GBJ002(GIL01) and its GIL01 cured derivative, GBJ002, treated or not with 100 ng mL$^{-1}$ of the DNA-damaging agent mitomycin C (MMC). We allowed the strain to develop as far as the early exponential growth phase, after which the cultures were split in two and MMC was added to one half of the culture to induce the SOS response to DNA damage. Samples for RNA sequencing (RNA-seq) were taken 30 min after MMC addition, which corresponds to the time just before the burst of GIL01 virions[20,21]. Gene-expression profiles were then determined using differential expression analysis (see Methods).

In strain GBJ002, 26.7% (1166 chromosomal genes, 240 pBtic235 genes) of the genes showed changes in expression levels after treatment with MMC (Fig. 1a; Supplementary Data 1). As expected, there was an increase in expression of the core SOS-response genes *lexA* (2.6-fold), *recA* (4.6-fold), *dinB* (DNA polymerase IV, 4.9-fold), and *uvrABC* (exonuclease components, 2.4- to 11.3-fold), and of several other genes involved in the transposition or upkeep of genome integrity, such as a transposase (3.8-fold) and topoisomerase IV genes A and B (6.3-fold and 6.2-fold, respectively). Interestingly, 235 of the 287 (82%) differentially expressed pBtic235 genes were upregulated (Supplementary Data 1). Consistent with a previous report showing the induction of prophage pBtic235 after MMC treatment[22], the upregulated genes included 68 genes encoded on the anti-sense strand of pBtic235, corresponding to the phage module including the phage capsid proteins and lytic genes, whereas the remaining 172 genes were encoded on the plasmid module. These included a RecA homolog (23.8-fold) that shares 42% amino acid identity with the host RecA, a LigA homolog (3.6-fold), a UvrD-helicase-like protein (46.9-fold), a crossover junction endodeoxyribonuclease RuvC (7.9-fold), 18 tRNA genes (10.9- to 61.8-fold), and a replication initiator protein A (RepA) (23.7-fold), indicating the involvement of pBtic235 in maintaining genome DNA integrity and replication of plasmid pBtic235 when the host is exposed to genotoxic conditions.

To determine whether prophage GIL01 affects the host and pBtic235 transcription we performed a differential expression analysis between strains GBJ002(GIL01) and GBJ002. We found that GIL01 triggered extensive changes in the transcriptome of GBJ002(GIL01), as 7.4% of the 5267 genes were differentially

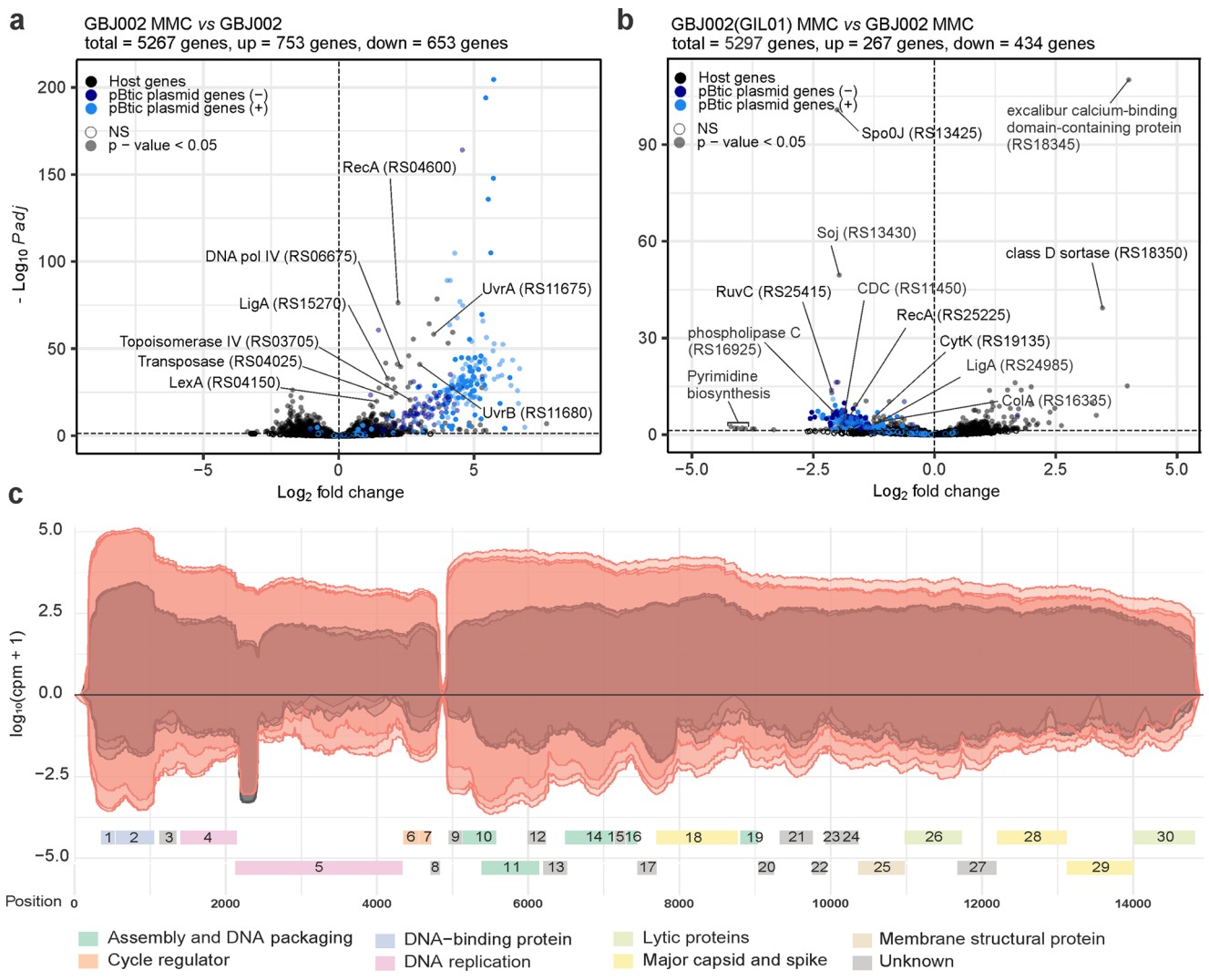

**Fig. 1 *Bacillus thuringiensis* response to mitomycin C (MMC) treatment and prophage GILO1 induction.** Volcano plots showing gene-expression changes between: **a** MMC-treated (100 ng mL$^{-1}$, 30 min) and untreated GBJ002 cultures; **b** MMC-treated (100 ng mL$^{-1}$, 30 min) GBJ002(GIL01) and GBJ002 cultures. The pBtic235 genes encoded on the sense (+, plasmid module) and anti-sense (−, phage module) strands, and the host genes are shown in the indicated colors. A significance cut-off of a FDR-adjusted *p*-value (Padj) of ≤0.01 was used. Genes without statistically significant fold-changes are indicated as NS. CDC, cholesterol-dependent anthrolysin O/cereolysin O family member protein and ColA, collagenase. Experiments were performed in three biological replicates. **c** GIL01 transcriptome in strain GBJ002(GIL01) 30 min after MMC treatment. The graph represents the GIL01 genome (ORF1-ORF30) with normalized read coverage for three biological replicates treated with MMC (100 ng mL$^{-1}$) (read coverage in red) compared to their respective untreated samples (read coverage in gray). Reads are represented on a logarithmic scale (log$_{10}$ counts per million). Gene features are colored based on putative or known gene functions[15]. Experiments were performed in three biological replicates.

expressed in non-induced cells and 12.7% in cells induced by DNA damage (Fig. 1b, Supplementary Data 2 and 3). The presence of GIL01 increased the expression of 237 and 123 and decreased the expression of 229 and 134 bacterial genes in the MMC-treated and untreated cells, respectively. In both cases most of the genes with increased expression had no annotated function or the genes could not be grouped according to gene ontology. In response to DNA damage, we detected the induction of all GIL01 prophage determinants (ORF1-ORF30) except for ORF8 (Supplementary Data 2).

Among the most down-regulated genes in presence of GIL01 prophage and induced DNA damage were the determinants involved in various functions, such as: pyrimidine biosynthesis (*e.g., pyrB* encoding aspartate carbamoyltransferase, *pyrK* and *pyrD* encoding subunits of dihydroorotate oxidase, *pyrC* encoding dihydroorotase, *carB* encoding a subunit of carbamoyl-phosphate synthetase; 18.4- to 9.9-fold), the transport of substrates through

the cell wall (*e.g.*, ABC transporter permeases, PTS transporter all without annotated gene names; 5.5- to 4.3-fold), virulence (e.g., M4 family metallopeptidase, *plc* encoding phospholipase C, cholesterol-dependent anthrolysin O/cereolysin O family member, *nheA*, *nheB*, and *nheC* encoding nonhemolytic enterotoxin, *cytK* encoding hemolysin IV; 4.3- to 2.0-fold) and sporulation (e.g. *spo0J* encoding stage 0 sporulation protein, *soj* encoding sporulation initiation inhibitor protein, 5.0- to 2.9-fold), among others (Fig. 1b, Supplementary Data 2). A similar profile of silenced genes was also observed in the GIL01 and pBtic235 lysogen, GBJ002(GIL01), which grew under non-stressed conditions, supporting lysogenic growth (Supplementary Data 3). Remarkably, the GIL01 prophage decreased the expression of more than 130 genes of the pBtic235 plasmid and phage modules under both conditions (Fig. 1b, Supplementary Data 2 and 3), suggesting that the GIL01 prophage inhibits the induction of the cohabiting pBtic235 prophage.

**Table 1 Enumeration of phage progeny released from strains GBJ002(GIL01), GBJ002(pDG) and GBJ002(pDG7).**

| Time after MMC induction | | | | |
| --- | --- | --- | --- | --- |
| Strain | 0 min | 30 min | 1 h | 4 h |
| GBJ002(GIL01) | $8.3 \times 10^6 \pm 2.9 \times 10^6$ | $4.3 \times 10^7 \pm 4.7 \times 10^6$ | $1.7 \times 10^{10} \pm 9.0 \times 10^9$ | NA |
| GBJ002(GIL01)* | $1.1 \times 10^7 \pm 3.9 \times 10^6$ | $3.3 \times 10^7 \pm 9.4 \times 10^6$ | $1.9 \times 10^{10} \pm 1.5 \times 10^{10}$ | NA |
| GBJ002(pDG) | $7.5 \times 10^4 \pm 5.0 \times 10^3$ | $1.5 \times 10^5 \pm 5.5 \times 10^4$ | $2.7 \times 10^5 \pm 1.1 \times 10^5$ | $1.8 \times 10^6 \pm 5.5 \times 10^5$ |
| GBJ002(pDG7) | $1.7 \times 10^4 \pm 5.0 \times 10^2$ | $3.5 \times 10^4 \pm 1.5 \times 10^4$ | $2.0 \times 10^4 \pm 5.0 \times 10^2$ | $1.5 \times 10^3 \pm 5.0 \times 10^2$ |
| GBJ002(pDG)* | 0 | 0 | 0 | 0 |

Strain GBJ002(GIL01) contains both the GIL01 and pBtic235 prophages whereas, strains GBJ002(pDG) and GBJ002(pDG7) carry only the pBtic235 prophage. The lytic cycle of both prophages was induced by adding MMC (100 ng mL$^{-1}$) to *Bacillus* cultures in the early exponential phase. Dilutions of the culture supernatants taken at 0 min, 30 min, 1 h and 4 h after MMC induction were spotted onto lawns of recipient strain HER1410 and the plaques formed were counted (PFU mL$^{-1}$) to determine the number of phage progeny produced after the different times. An asterisk (*) indicates samples vortexed for 10 s. Not assessed (NA).

The transcriptional landscape of GIL01 confirmed the presence of two main transcription modules, which were both upregulated in response to the DNA damage (Fig. 1c). The upstream module encompassed genes encoding for gp1-gp8, which are responsible for genome replication and transcription regulation. Thirty minutes into the DNA-damage response (Supplementary Data 4), 55.9-fold more sequencing reads were assigned to this module compared to the untreated samples (Fig. 1c). Interestingly, in the absence of MMC, most reads assigned to the GIL01 anti-sense strand matched ~230 nucleotides near the 5'-end of the DNA polymerase gene (Fig. 1c). An analysis of the transcript in all three reading frames revealed no ORF with a length of >35 amino acids, which suggested that this region was non-coding. As this transcript levels decreased almost in proportion to the accumulation of transcripts for the two genes involved in the GIL01 genome replication in the DNA-damage response (i.e., gp4, gp5), this suggested an anti-sense interference mechanism of action. Further work is required to confirm the biological role of this highly abundant transcript.

According to our transcriptomic data the downstream 10-kb module of GIL01 encompasses genes 9 to 30, which encode the structural, lytic, and host recognition functions. Interestingly, both modules showed relatively high expression levels in the cells not treated with MMC. As the RNA-seq data in Fig. 1c represent the mean gene expression of the bacterial population, the apparent high expression of the GIL01 genes might reflect a high, spontaneous prophage induction within a portion of the population.

To confirm that spontaneous induction is indeed an intrinsic property of GIL01[13], the number of plaque-forming units (PFUs) on the recipient host *B. thuringiensis* HER1410[23] was determined in a supernatant of GBJ002(GIL01) cultures treated or not with MMC during the exponential growth phase. These showed titers of approximately $10^7$ and $10^{10}$ PFU mL$^{-1}$, just before and one hour after the induction of the host SOS response (Table 1). It is worth noting that the pBtic235 virions, but not the GIL01 virions, are inactivated by vortexing the phage samples (Table 1). This allowed us to inactivate only phage pBtic235 in the samples. Our results indicate a high frequency of spontaneous induction of GIL01, which explains the high expression of the structural and lysis determinants of GIL01 in bacteria without chemical induction (Fig. 1c).

**Gp7 regulates host gene expression and represses pBtic235 genes after DNA damage.** In previous work we established that GIL01 protein gp7 is needed for lysogeny and exerts its co-repressor function by interacting with the host repressor LexA[16,17]. Here, we investigated the effects of gp7 on the host and pBtic235 transcriptional programs. For this we transformed strain GBJ002 with a vector for the IPTG-induced expression of gp7 [GBJ002(pDG7)] or its empty version [GBJ002(pDG)]. The strains were treated with MMC to induce the SOS response to DNA damage and prophage pBtic235[22]. RNA-seq and differential expression analysis were then used to profile the gene expression, both before and 1 h after the DNA damage. It is worth noting that quantitative PCR analysis (Supplementary Methods) shows that gp7 mRNA levels are comparable when gp7 is expressed from the IPTG-induced pDG7 plasmid or from the MMC-induced GIL01 prophage carried by the GBJ002(GIL01) strain (Supplementary Fig. 2). Differential expression analysis between the MMC-treated strains GBJ002(pDG7) and GBJ002(pDG) revealed substantial changes across the *B. thuringiensis* transcriptome, especially within the pBtic235 phage module. Gp7 affected the expression of 58 chromosomal genes and 226 pBtic235 genes (Fig. 2a; Supplementary Data 5). It predominantly down-regulated gene expression, repressing several host core SOS genes (e.g., *recA*, *uvrAB*, *dinB*, by ~2-fold) and genes involved in cell-wall transport and integrity, as well as in protein and nucleic acid hydrolysis (Supplementary Data 5). Interestingly, among the few genes that gp7 upregulated in the stressed cells there were several virulence factors[24]: the two pore-forming cytolysins hemolysin IV (CytK) by 2.3-fold, the cholesterol-dependent anthrolysin O/cereolysin O family member by 3.3-fold (GenBank ID, CP051858; locus tags, RS19135, RS11450, respectively), collagenase ColA by 2.5-fold, and phospholipase C by 2.3-fold (locus tags, RS16335, RS16925, respectively). This finding suggests that gp7 induced the virulent properties of *B. thuringiensis* during stress (Fig. 2a; Supplementary Data 5). However, the data contrast with the results showing that the GIL01 prophage down-regulates these virulence determinants (Fig. 1b, Supplementary Data 2), indicating that other unknown regulatory mechanisms are at play.

A comparison of the pBtic235 transcriptome landscape in the MMC-treated GBJ002(pDG7) and GBJ002(pDG) strains shows that gp7 repressed gene expression of both the plasmid and the phage module of pBtic235 (Fig. 2a). For most differentially expressed pBtic235 genes, the functions were not annotated and genes of known functions are mostly involved in protein synthesis and maintenance of genome integrity (Supplementary Data 5).

We wanted to understand the molecular mechanisms by which gp7 coordinates the induction of pBtic235. First, we asked whether pBtic235 is regulated by LexA, and screened regions 300 bp upstream of pBtic235 genes for sequence motifs within promoter regions consistent with the LexA consensus motif of *B. subtilis*[25] using xFITOM[26]. The search revealed a putative LexA binding sequence located between coordinates 73794 and 73807 in the pBtic235 sequence (GenBank ID, NZ_CP051859) (Fig. 2b, Supplementary Fig. 1). Next, we investigated the interaction of purified *B. thuringiensis* LexA with a DNA fragment carrying the putative LexA binding sequence identified using surface plasmon resonance (SPR) spectrometry. As we have previously shown that gp7 enhances LexA binding to the GIL01 genome and host SOS promoter sites[16], we tested whether gp7 favored LexA binding to

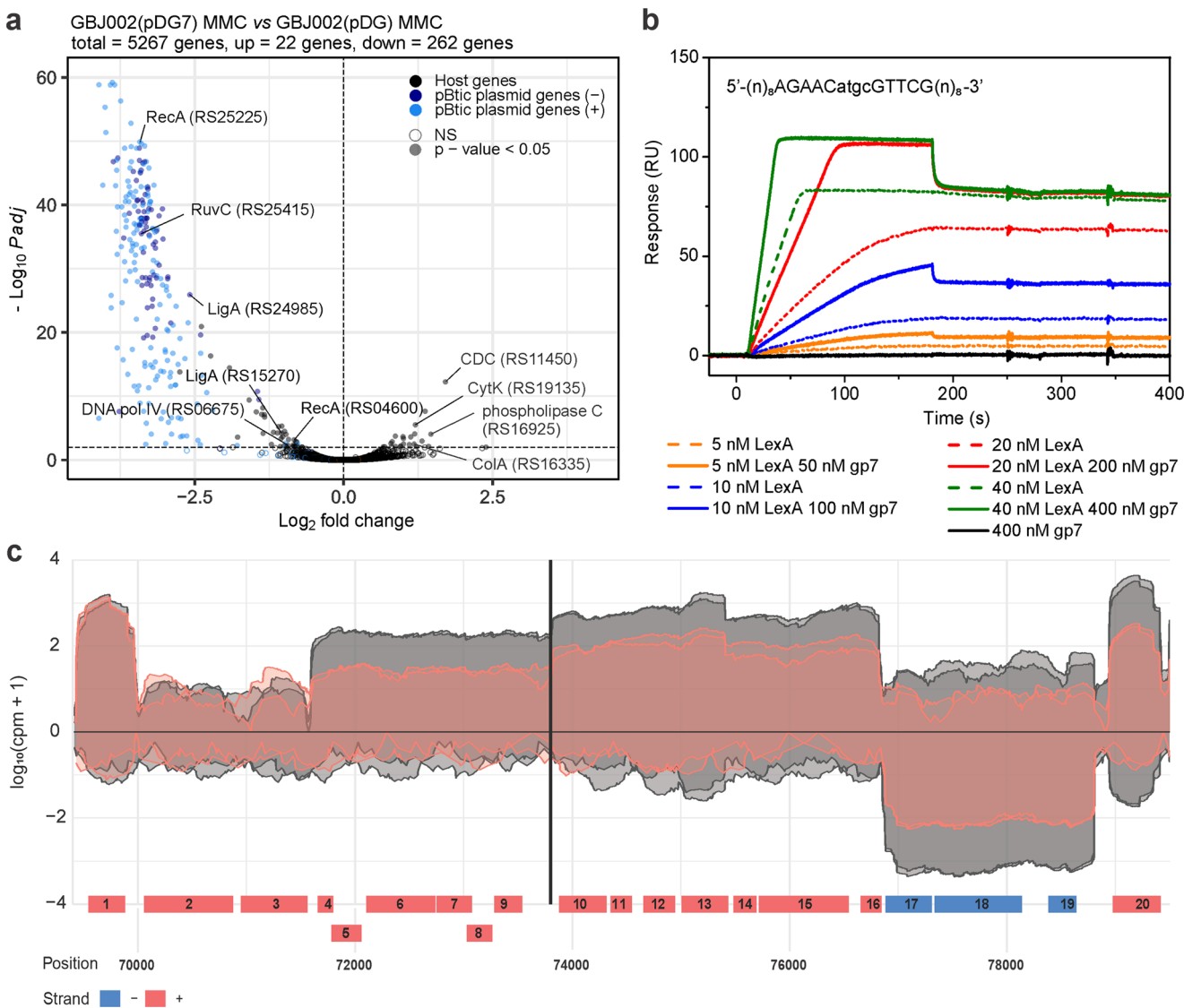

**Fig. 2 Gp7 interacts with host LexA to repress the transcription of pBtic235. a** Volcano plot showing the changes in gene expression between MMC-treated (100 ng mL$^{-1}$, 1 h) and IPTG-induced (0.1 mM) GBJ002(pDG7) and GBJ002(pDG) cultures. The pBtic235 genes encoded on the sense (+, plasmid module) and anti-sense (−, phage module) strands, and the host genes are shown in the indicated colors. A significance cut-off of a FDR-adjusted p-value (Padj) of ≤0.01 was used. Genes without statistically significant fold-changes are indicated as NS. CDC, cholesterol-dependent anthrolysin O/cereolysin O family member protein and ColA, collagenase. Experiments were performed in two biological replicates. **b** Surface plasmon resonance sensorgrams of LexA (5–40 nM) interactions in the presence (solid lines) and absence (dotted lines) of gp7 (50–400 nM), with ~40 response units (RUs) of chip-immobilized LexA operator (sequence indicated above sensorgrams; pBtic235 coordinates 73794 to 73807) in pBtic235 (sequence above the diagram). Gp7 (400 nM) alone does not interact with the LexA operator (black line). Experiments were performed in three technical replicates and the represenatative sensorgrams are shown. **c** Read coverage (log$_{10}$ counts per million) of the pBtic235 genomic region (from 70 to ~79 kb) flanking the LexA operator site (black vertical line) 1 h after MMC treatment (100 ng mL$^{-1}$) in strain GBJ002(pDG7) with ectopically expressed gp7 (read coverage in red) and in strain GBJ002(pDG) without gp7 (read coverage in gray). The experiment was performed in two biological replicates. Genes labeled 1–20 correspond to the pBtic235 genes HIS92_RS25435 to HIS92_RS25530, respectively. Most lack functional annotation, except for the genes labeled 2 (HIS92_RS25440; phage repressor protein/anti-repressor Ant) 13 (HIS92_RS25495; nucleoside 2-deoxyribosyltransferase) 15 (HIS92_RS25505; metallophosphoesterase) and 18 (HIS92_RS25520; PD-(D/E)XK nuclease family protein).

the putative target site on pBtic235. Our SPR data showed that LexA bound to the target site on pBtic235 and that gp7 enhanced LexA binding to the DNA, but did not interact directly with the LexA operator (Fig. 2b).

The identified LexA target site was located upstream of a putative operon that consist of seven ORFs (gene IDs, RS25480 to RS25510), one of which showed homology with the nucleoside 2-deoxyribosyltransferase (RS25495) and another with metallo-phosphoesterase (RS25505). The sequences of the remaining ORFs did not match any genes with known functions. A closer

look at the transcriptomic data for GBJ002(pDG7) (Fig. 2a, Supplementary Data 5) revealed that ORFs RS25480 to RS25510 were repressed by gp7 (Fig. 2c). Together, the SPR spectrometry and transcriptomic data suggested that these ORFs were organized into an operon that is repressed by LexA and its co-repressor gp7.

In presence of gp7, we also observed the repression of genes RS25450 to RS25475 and RS25515 to RS25530 (Fig. 2c) despite the absence of a detectable LexA binding site upstream these genes or their presumed operons. It is therefore possible that

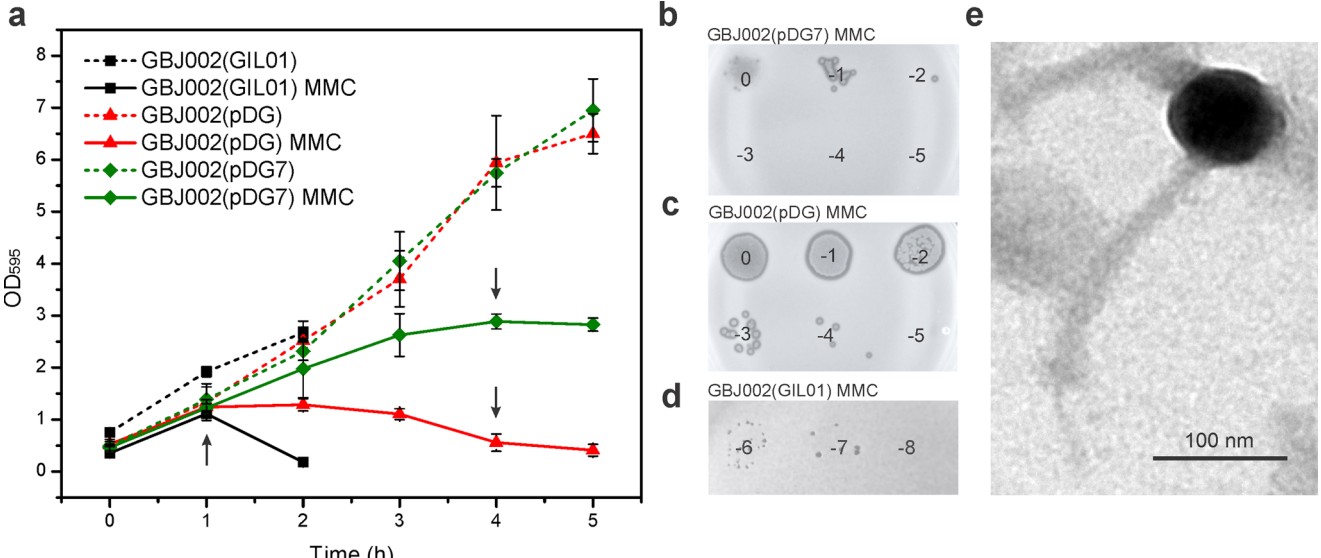

**Fig. 3 Gp7 represses the induction of the lytic cycle of jumbo phage pBtic235. a** Comparison of growth kinetics of *B. thuringiensis* strains GBJ002(GIL01) (black), GBJ002(pDG) (red), and GBJ002(pDG7) (green). IPTG (0.1 mM) was added to GBJ002(pDG) and GBJ002(pDG7) cultures after dilution of the overnight culture in fresh LB medium. MMC (100 ng mL$^{-1}$) was added to half of the cultures in the early exponential phase (OD$_{595}$, 0.3–0.4) at time zero. Bacterial density of the MMC-treated cultures is shown with solid lines, and that of the untreated cultures is shown with dotted lines. Bacterial growth was measured for 5 h in 1-h intervals or until complete cell lysis of MMC-treated GBJ002(pDG) culture. Black arrows indicate the times at which the culture samples were collected for determination of phage counts in **b–d**. Each data point represents the mean of three replicates. Error bars indicate standard deviations. Spot titers of MMC-treated culture supernatants of strains GBJ002(pDG7) **b**, GBJ002(pDG) **c** and GBJ002(GIL01) **d** on the recipient host, *B. thuringiensis* HER1410. The experiment was performed in duplicate. **e** Representative transmission electron micrograph of the pBtic235 phage showing an isometric head of ~80-nm diameter and an ~200-nm-long tail. Scale bar, 100 nm.

ORFs RS25480 to RS25510 encode a regulator controlling the expression of those genes or that gp7 enables LexA binding to poorly conserved sites, as observed in phage GIL01[16].

**Gp7 represses the induction of the jumbo phage pBtic235.** Our transcriptomic and SPR data raised the question whether gp7 coordinates the cross-talk between the transcriptional programs of GIL01 and pBtic235, to orchestrate the induction of both prophages in *B. thuringiensis*. To test this, we compared the growth dynamics of strain GBJ002(GIL01) and the pBtic235 lysogen cured of GIL01 (GBJ002), the latter with and without IPTG-induced gp7. First, we determined the level of spontaneous induction of the pBtic235 prophage by analyzing the formation of phage plaques on the strain HER1410[22]. In the exponential growth phase of the GBJ002(pDG) strain, we detected a spontaneous induction of pBtic235 prophage of 7.5 ×10$^4$ PFU mL$^{-1}$ (Table 1). At 4 h after the induction of the DNA damage, the number of pBtic235 infectious particles increased 240-fold. Next, we determined the extent of gp7 on pBtic235 virion production. Gp7 did not have detectable effects on cell growth under normal conditions, but after 4 h of inducing DNA damage, gp7 protected the cells from pBtic235-induced lysis (Fig. 3a), and we measured by up to 1000-fold reduction in pBtic235 plaque-forming units released in the medium of GBJ002 expressing gp7 (Fig. 3b, c and Table 1, compare GBJ002(pDG7) and GBJ002(pDG)). In contrast, the GBJ002(GIL01) strain carrying both the GIL01 and pBtic235 prophages, lysed at the fastest rate, as its culture was completely lysed 2 h after MMC treatment (Fig. 3a, d). Consistent with the observed down-regulation of pBtic235 prophage genes by the GIL01 prophage or by ectopically expressed gp7 (Figs. 1b and 2a), these data show that gp7 of phage GIL01 inhibits the production of pBtic235 virions under DNA-damaging conditions. To better investigate the role of LexA in the induction of prophage pBtic235, we used a GBJ002 derivative strain GBJ396

carrying a gene for LexA repressor that is incapable of self-cleavage[14]. We sequenced strains GBJ002 (SRA ID: SAMN30884992) and GBJ396 (SRA ID: SAMN30884991) and alignment of the sequencing reads revealed that the GBJ396 strain lacks the pBtic235 prophage. Using the soft agar overlay method with the GBJ396 strain as host, we observed that pBtic235 was unable to infect GBJ396.

To further characterize the phage pBtic235 we induced its lytic cycle in strain GBJ002 with MMC, and generated pBtic235 plaques on the recipient strain. Although pBtic235 mostly formed turbid plaques, there was a single clear plaque, which suggested a lytic phage variant that could not establish the dormant state in the cell. We performed several rounds of propagation with the clear-plaque variant during repeated infections of HER1410, and extracted its DNA from the confluent plaques for sequencing. We mapped the reads to the published pBtic235 genome, which confirmed that the lytic plaques contained the pBtic235 genome[22]. The largest assembled contig sequence of 225.5 kb aligned with the deposited 235 kb pBtic235 sequence[22] (GenBank ID, NZ_CP051859), with the exception of a single nucleotide change that resulted in an alanine-to-valine substitution of residue 925 of the minor structural protein carrying a carbohydrate-binding domain (NCBI Protein ID, AFQ30225). Remarkably, the read assembly revealed a ~10-kb deletion in the clear-plaque variant, which encompassed 12 genes (Supplementary Fig. 3), including *repA*, which indicated that pBtic235 genome replication can initiate without RepA. We detected multiple reads that connected the ends of the deleted region, showing that like the wild-type pBtic235 genome, the mutant pBtic235 genome was also circular[19]. Next, we purified the pBtic235 virions from the MMC-induced culture of GBJ002 and visualized their morphology using transmission electron microscopy. The micrographs showed that the phage pBtic235 belongs to the order *Caudovirales*, as it was composed of an isometric

head of ~80-nm in diameter (edge-to-edge) and a tail of ~12 nm in width and ~200 nm in length, with a central tail fiber or spike at the base plate (Fig. 3e).

**Tectivirus-borne gp7 homologs enhance the DNA-binding activity of LexA.** Considering the global regulatory role of gp7 in *B. thuringiensis* we investigated the existence of gp7-like proteins in other bacteria. In a previous study we established that gp7 can interact with the LexA of *Staphylococcus aureus* and enhance its DNA binding[17]. Therefore, proteins with similar functional properties to gp7 might modulate the LexA activity in *S. aureus*. This might also be the case in *Escherichia coli*, where an unidentified factor appears to be required for LexA binding to multiple target sites[27]. These observations prompted us to search for LexA co-regulatory proteins in *S. aureus* and *E. coli* using affinity chromatography and mass spectrometry (Supplementary Methods), with *S. aureus* and *E. coli* LexA as bait, and the respective cleared cell extracts as the sources of potential gp7-like proteins (Supplementary Data 6). In addition, we applied the ProBiS algorithm[28] to search the SWISS-MODEL Repository for proteins of *E. coli*, *S. aureus*, and *B. thuringiensis* that consist of 40–100 amino acids and share surface structural features with gp7. Using these approaches we were unable to identify a gp7-like protein (Supplementary Fig. 4).

We previously identified gp7 homologs exclusively in genomes of GIL01-related tectiviruses infecting *Bacillus* spp.[29], although it is not known whether these have a similar role by interacting with LexA. Here, we purified six gp7 homologs from phages infecting insect, fish, and human pathogens *Bacillus thuringiensis* serovar *aizawai*, *Bacillus mycoides*, *Bacillus anthracis*, and *Bacillus cereus* (Supplementary Figs. 5 and 6). The sequence alignment of these gp7 homologs showed that they mainly differed in their C-terminal domain (Fig. 4a), which we previously found to be an unstructured region with at least the last six C-terminal residues not being required for the formation of the LexA complex[17]. To test whether the six gp7 homologs can form a complex with LexA we used SPR to measure the protein-protein interactions. *B. thuringiensis* LexA was immobilized on the surface of the SPR sensor chip, and gp7 or one of its homologs was injected over the chip. All six gp7 homologs interacted with the *B. thuringiensis* serovar *israelensis* LexA, and this interaction was concentration dependent, with $K_D$s similar to that of gp7, in the range of 148 nM to 274 nM (Fig. 4b). It is of note that the LexA repressors of *B. thuringiensis* serovar *aizawai* or *israelensis*, *B. mycoides*, *B. anthracis*, and *B. cereus* share >96% identity (Supplementary Fig. 7). To test the activity range of the gp7-like proteins beyond *Bacillus* spp., we exposed the six homologs to *S. aureus* and *E. coli* LexA and found that all of these interacted with *S. aureus* LexA ($K_D$ from ~300–790 nM) (Supplementary Fig. 8), although they did not significantly interact with the *E. coli* LexA (Supplementary Fig. 9). The *S. aureus* and *E. coli* LexA share 67% and 38% identity, respectively, with the *B. thuringiensis* serovar *israelensis* LexA, which indicated that LexA-sequence conservation is needed for gp7 or its homologs to interact with LexA. As previously seen for gp7, no gp7-like protein interacted directly with the LexA operators from the GIL01 genome (Supplementary Fig. 10), and all of the homologs enhanced *B. thuringiensis* LexA binding to its cognate operator sequence. It is of note that of the three LexA repressors tested, the LexA proteins from *S. aureus* and *B. thuringiensis* showed the highest sequence identities[17]. Moreover, all the gp7 homologs tested enhanced the interaction of *B. thuringiensis* LexA with the cognate operator sequence (Fig. 4c) and we would expect them to co-regulate the DNA-damage responses in their respective hosts.

## Discussion

The data provided here show that a hierarchy has evolved in *B. thuringiensis* serovar *israelensis* between the two co-residential prophages carried on the phage-plasmid extrachromosomal elements pBtic235 and pGIL01[30]. Transcriptomic analysis of the interaction between *B. thuringiensis* and the two prophages revealed that the linear 15-kb prophage GIL01 represses more than 71% of the plasmid and phage module genes carried by pBtic235 in the DNA damage response. Remarkably, we showed that GIL01 deploys gp7 to control the phage-pBtic235 cycle. The transcriptomic data show that ectopic expression of gp7 at native levels down-regulates the pBtic235 genes in the presence of a prophage-inducing agent. Gp7 significantly repressed the production of phage pBtic235 when GBJ002, the pBtic235 lysogen, was exposed to DNA-damaging stress. In contrast to its role in GIL01[14], we show here that gp7 is not crucial for establishing the phage-pBtic235 lysogenic cycle, and that the key determinants of pBtic235 required for quiescence appear to be located in the ~10-kb region, which is absent in a pBtic235 clear-plaque mutant. Thus, gp7 is a dual-function regulator that, depending on the environmental conditions, enables GIL01 to either enter the dormant state or be induced before the co-resident prophage pBtic235.

Gp7 acts as a master regulator as it down-regulated genes that constitute the core LexA-regulon in bacteria[25,31], with roles in SOS regulation, DNA recombination and excision, and error-prone DNA repair. Unexpectedly, gp7 upregulated the genes for two cytolysins, a phospholipase C, and a collagenase in SOS-induced cells, which suggests that gp7 can influence the synthesis, and possibly the release, of *B. thuringiensis* virulence determinants[32]. However, the phage GIL01 appears to have evolved mechanisms beyond the gp7 regulator. We observed that in response to DNA damage, the prophage GIL01 not only inhibited the expression of the pBtic235 genes but also repressed the transcription of the host virulence factors, but did not affect the expression of the SOS genes. We hypothesize that certain GIL01 prophage determinants have co-evolved with gp7 to preserve the sensitivity of the cellular DNA-damage response, which is important for host survival, and to fine-tune certain host functions, such as virulence-factor expression, with the lytic life cycle processes of GIL01.

Although most bacteria carry multiple prophages[33], the regulatory mechanisms that coordinate their induction at the transcription level remain largely unexplored. The food-borne pathogen *Listeria monocytogenes* has been shown to carry a co-regulated functional prophage and a cryptic phage element controlled by the same inducer[5]. A DNA-damage-inducible metalloprotease on the cryptic phage inactivates the cI-like repressors of both prophages, resulting in synchronized lytic induction. Interestingly, the induction of these prophage elements is attenuated by the prophage repressor AriS, which interferes with the host SOS response. However, the inhibitory effect of the SOS response was only observed when the AriS was ectopically overexpressed and not transcribed from the prophage[34]. This is consistent with our observation that in addition to gp7, other GIL01 factors also influence host gene expression. In addition, *Salmonella enterica* prophages Gifsy-1 and Gifsy-3 carry SOS-induced anti-repressors that also cause the simultaneous induction of prophages into the lytic cycle function by direct binding and the displacement of cognate and noncognate phage repressors bound at target DNA sites[4]. To the best of our knowledge, the present study is the first report of a small phage-borne protein that modulates transcriptional patterns in a cohabiting phage to extend its latency.

In *E. coli*, an as-yet-unidentified factor is required for LexA to bind noncanonical sequence motifs[27], and in the multidrug-

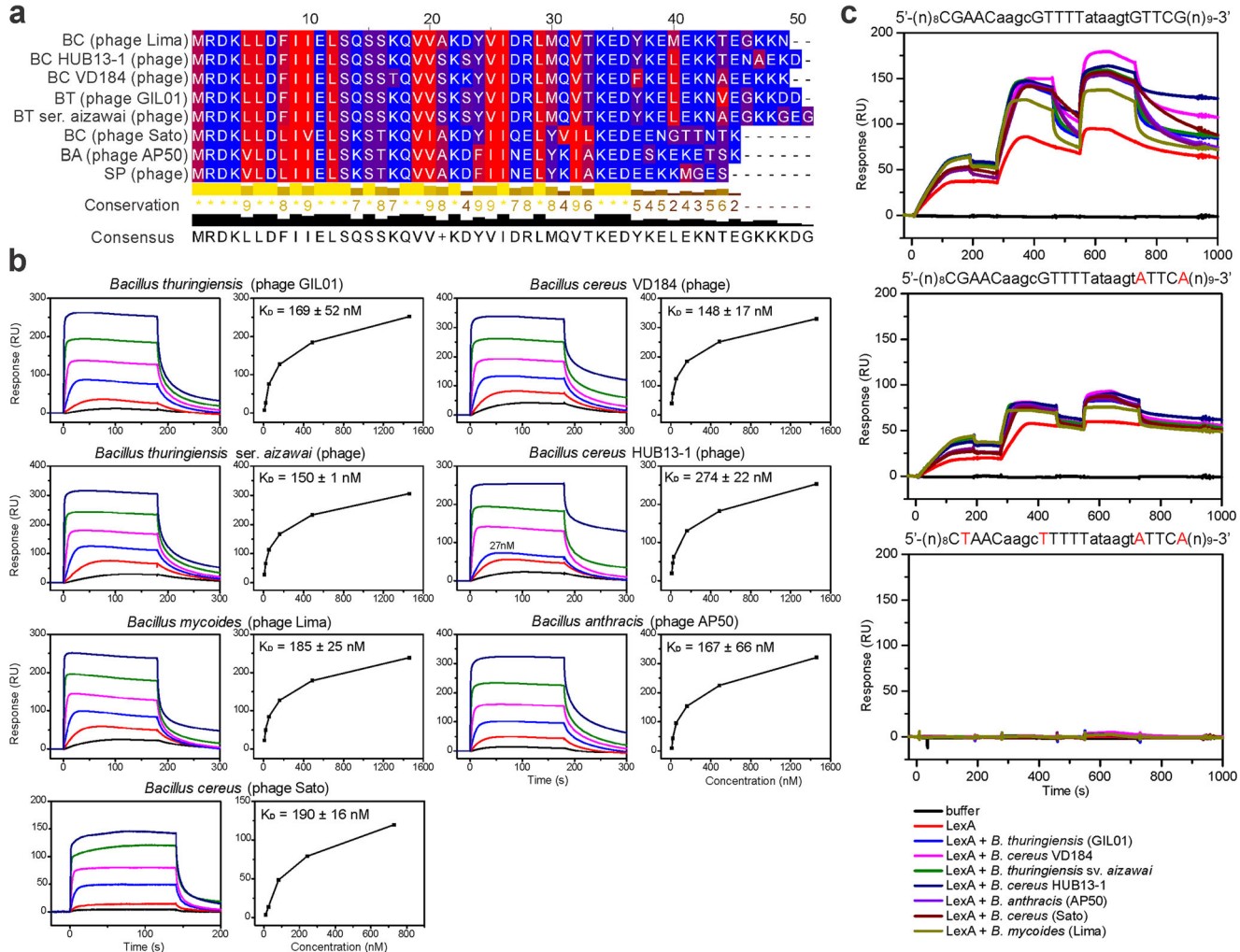

**Fig. 4 Tectiviral gp7 homologs interact with and enhance the DNA-binding properties of *B. thuringiensis* LexA. a** Amino-acid sequences of tectiviral gp7 homologs aligned with Clustal Omega and visualized using Jalview. Amino acids are color coded according to the hydrophobicity scale, with red indicating hydrophobic residues, and blue indicating hydrophilic residues. **b** Surface plasmon resonance sensorgrams of the interaction of gp7 and gp7 homologs (respective hosts are indicated above the diagrams) with immobilized *B. thuringiensis* LexA. Analytes were injected at 3-fold serial dilutions (6–1458 nM) for 180 s at a flow rate of 30 μL min$^{-1}$ over chip-immobilized LexA. Apparent equilibrium dissociation constants (K$_D$) are expressed as means ± standard deviations of two titrations of each analyte. **c** Surface plasmon resonance sensorgrams of the interaction between *B. thuringiensis* LexA alone or pre-incubated with gp7 or gp7 homologs with the wild-type or mutant LexA operators from the GIL01 *P1* promoter region[16]. Sequences of DNA fragments used are shown above the diagrams, with important nucleotides for LexA binding shown in capital letters, and changed nucleotides shown in red. Free LexA (5, 10, 20 nM) or LexA (5, 10, 20 nM) pre-incubated with gp7 and gp7-like proteins in 1:10 molar ratio (50, 100, 200 nM) were injected over the selected DNAs immobilized on the chip at 50 μL min$^{-1}$ in a single cycle. The experiment was performed in duplicate and representative sensorgrams are shown.

resistant pathogen *Acinetobacter baumanii*, the LexA-like repressor UmuDAb controls the DNA-damage-inducible genes through a direct interaction with the ~9-kDa co-repressor DdrR, although the mechanism is not yet fully understood[35,36]. Similar to the narrow distribution of gp7 homologs across the tectivirus genomes, DdrR is only found in the chromosomes of *Acinetobacter* spp.[37]. This suggests that the LexA accessory proteins have co-evolved with the LexA repressors of their hosts, and thus have different origins but function congruently. We anticipate that the present study will provide the impetus to explore the small gp7-like proteins of viral and bacterial origin that link the host DNA-damage responses and other processes to the prophage induction and the movement of co-resident mobile genetic elements between bacteria[21,38,39]. It is tempting to speculate that this 'know-how' hidden in the LexA accessory proteins might offer a way to manipulate pathogens.

## Material and methods

**Bacterial strains, plasmids, and culture conditions.** All the strains and plasmids used in this study are listed and described in Supplementary Table 1. The *B. thuringiensis* strains were routinely cultured in a Lysogeny Broth (LB) medium at 30 °C and 180 rpm, unless otherwise indicated. The *E. coli* and *S. aureus* strains were grown in a LB at 37 °C and 180 rpm. For overnight cultures, a single bacterial colony grown on LB agar plates was inoculated into 5 mL of LB medium and incubated for 14 h to 16 h at 30 °C and 180 rpm. Exponential cultures were prepared by inoculating the overnight culture into a fresh medium in a ratio of 1:100. Kanamycin (Sigma) was added to the culture medium at 50 μg mL$^{-1}$ for solid LB agar plates at 25 μg mL$^{-1}$ for overnight cultures and 10 μg mL$^{-1}$ for exponential cultures to select for plasmids pDG and pDG7. For the agar plates, the LB medium was solidified with 1.5% (w/v) agar and soft-agar overlays with 0.4% (w/v) agar. Soft agar overlays containing the strain *B.*

*thuringiensis* HER1410 as a recipient host were prepared by mixing 200 μL of the culture in an early exponential phase with 5 mL of soft agar at 48 °C. The soft-agar mixture was then poured onto warm LB agar plates and left to solidify for 30 min at room temperature before spotting 10 μL serial dilutions of the phage stocks. Protein expression from the pDG7 plasmid was induced by adding 0.1 mM isopropyl-β-D-1 thiogalactopyranoside (IPTG; final concentration) to the cultures.

**Sample preparation for RNA extraction and sequencing**. Cultures for the transcriptome analysis were prepared by inoculating 150 mL of LB with an overnight culture of GBJ002, GBJ002(GIL01), GBJ002(pDG) or GBJ002(pDG7) in the ratio 1:100 and the addition of 0.1 mM IPTG to the GBJ002(pDG) and GBJ002(pDG7) strains. The cultures were grown aerobically at 30 °C for approximately 3 h, until an early exponential phase (OD$_{595}$, 0.3-0.4). At this time, the cultures were divided into two and 100 ng mL$^{-1}$ of MMC was added to one of the cultures. Culture samples (7 mL) of MMC-treated or untreated GBJ002 and GBJ002(GIL01) were taken 30 min after the addition of MMC, and of GBJ002(pDG) and GBJ002(pDG7) 1 h after the addition of MMC. Immediately after collection the culture samples were centrifuged at 15,000× *g* for 90 s and resuspended in 800 μL of DNA/RNA Shield reagent (Zymo Research, USA), which preserves the nucleic acids at room temperature. Samples were then transferred to lysis tubing (0.1, 0.5 mm; ZR Bashing-Bead; Zymo Research) and mechanically homogenized by bead beating using a mixer mill (MM 301; Retsch) for 10 min at 30 Hz. The total RNA was isolated using ZymoBIOMICS RNA miniprep kits (Zymo Research) according to the manufacturer's instructions and then stored at −80 °C. The sequencing libraries and RNA sequencing were performed by Novogene (China). The sequencing library was constructed as follows. First, the ribosomal RNA was removed using an rRNA removal kit. The rRNA-depleted RNA was then fragmented and the first cDNA was synthesized using hexamer primers. Then the second cDNA strand was synthesized and the dUTPs in the reaction buffer were replaced with dTTPs. The directional library was ready after final repair, A-tailing, adapter ligation, size selection, USER enzyme digestion, amplification and purification. All of the experiments were performed as two biological replicates.

**Pipeline for RNA-seq analysis**. Samples were initially processed using a custom Snakemake[40] workflow (available at https://github.com/AnzeLovse/mag). We also distributed a ready-to-use analysis environment in the form of a Docker container, which is documented in the GitHub repository. First, adapter and quality trimming was performed using Cutadapt (2.10)[41] with the following parameters: quality base, 10,10; error rate, 0.1; minimum length, 1; and max n, 0.1. The adapter sequence AGATCG-GAAGAGCACACGTCTGAACTCCAGTCA was removed from the 3'-end of the forward reads, and AGATCGGAA-GAGCGTCGTGTAGGGAAAGAGTGT from the 3'-ends of the reverse reads. Next, Bowtie2 (2.4.1)[42] was used to map the reads on a combined reference genome that consisted of *B. thuringiensis* serovar *israelensis* strain BGSC 4Q7rifR chromosome (NCBI RefSeq, NZ_CP051858.1), the pBtic235 plasmid (NCBI RefSeq, NZ_CP051859.1), and the GIL01 phage (GenBank ID, AJ536073.2), for mapping the transcriptome dynamics samples. The very sensitive option of pre-set parameters (–very-sensitive) was used for the alignment. The gene expression was quantified using featureCounts (2.0.1)[43], where we specified reversely stranded (-s 2), paired-end reads (-p), and gene-level quantification (-t gene). Two factors of variation were removed from the GBJ002(pDG) and GBJ002(pDG7) samples using the RUVs

method from the RUVSeq R/Bioconductor package (1.30.0)[44] 300 genes that had no differential expression in the two samples (false discovery rate [FDR] < 0.1) were used in the RUVs procedure. Differential expression analysis was performed using the DESeq2 (1.36.0)[45] R/Bioconductor package. Before the analysis, non-expressed genes were filtered out by requiring more than five reads per gene in at least two samples. For the GBJ002(pDG) and GBJ002(pDG7) samples, pairwise comparisons were carried out using an α level of 0.01. Log fold changes were shrunken using the apeglm (1.18.0) package[46] and genes with FDR < 0.01 were considered differentially expressed. The results were visualized using the EnhancedVolcano (1.14.0) R/Bioconductor package.

For the differential gene-expression analysis, a likelihood-ratio test was performed first, where ~condition+time+condition:time was used for the full design, and ~condition+time was used for the reduced design. The α parameter was set to 0.05, and the genes with FDR < 0.05 were considered to be differentially expressed. Log fold changes of the significant genes were extracted from the interaction terms, and plotted using the ComplexHeatmap (2.12.0)[47] R/Bioconductor package. The heatmap rows (genes) were clustered using the complete method on Euclidean distances.

The genome coverage was calculated for each strand separately using the genomeCoverageBed command from Bedtools (2.30.0)[48], with the -bga, -strand, and -pc arguments. Coverage was then normalized to counts per million, log transformed, and visualized using the ggplot2 R package (3.3.6)[49].

Differential expression analysis of samples following prophage GIL01 induction was caried out in a similar manner. Briefly reads were filtered and analyzed with DESeq2. We used ~0 + condition for the experimental design. Genes with FDR < 0.05 are differentially expressed. The differential expression analysis code is available on GitHub (https://github.com/AnzeLovse/pavlin-lovse-2022).

**Analysis of the effect of ectopically expressed gp7 on the induction of prophage pBtic235**. Overnight cultures of the strains GBJ002(GIL01), GBJ002(pDG) and GBJ002(pDG7) were diluted 100-fold in 20 mL of LB and 0.1 mM IPTG was added to the GBJ002(pDG) and GBJ002(pDG7) cultures. The cultures were grown at 30 °C and 180 rpm to an early exponential phase (OD$_{595}$ 0.3–0.6), at which point the first OD$_{595}$ measurement was recorded and 100 ng mL$^{-1}$ MMC was added to half of the cultures. Subsequently, measurements were taken every hour until OD$_{595}$ dropped below 0.2. To determine the GIL01 and pBtic235 progeny production in response to MMC induction, 0.5 mL culture samples were taken before, 30 min, and 1 h after the addition of the MMC. Another sample of cultures GBJ002(pDG) and GBJ002(pDG7) was taken 4 h after the MMC's induction, when complete lysis was observed. The samples were centrifuged immediately after collection at 15,000 × *g* for 3 min, and a 100 μL of supernatant was stored separately. The supernatants were serially diluted in fresh LB by gentle flicking and 10 μL of each dilution was spotted onto a soft-agar overlay containing the recipient host HER1410. The same dilutions were then vortexed for 10 s to inactivate the phage pBtic235 and again spotted onto soft agar containing HER1410. The plates were incubated overnight at 30 °C and the plaque-forming units were enumerated to determine phage progeny production. The experiment was performed in three biological replicates.

**Purification of the pBtic235 phage and genome isolation, sequencing, and assembly**. The phage pBtic235 was induced from strain GBJ002 in the early exponential growth phase by the addition of 100 ng mL$^{-1}$ of MMC (final concentration) for 4 h.

The lysed culture (1 mL) was centrifuged at $6000 \times g$ for 5 min to remove the bacterial cells and cell debris. Serial dilutions (10 µL) of the supernatant were spotted onto a soft-agar overlay of the recipient strain HER1410 and incubated overnight at 30 °C. A selected clear plaque was purified through three consecutive infectious rounds of the strain HER1410. Next, a single clear plaque was picked, resuspended in 100 µL of LB and incubated with 200 µL of HER1410 in the early exponential growth phase at room temperature for 5 min. It was then poured into soft agar on an LB plate. The resulting confluent lawn of plaques was scraped into a 50-mL tube (Falcon) and the plate was rinsed with 5 mL of sterile phosphate-buffered saline, which was transferred to the tube along with the soft-agar lysate. To release the phage particles from the soft agar into the buffer, the tube was mixed end-over-end for 10 min at room temperature. The phage lysate was centrifuged at $6000 \times g$ for 15 min. The phage DNA was extracted as previously described[13] and sequenced by Eurofins (UK). The genome coverage was calculated using the genomeCoverageBed command from Bedtools (2.30.0) with the -bga argument, and the pBtic235 genome as a reference (GenBank ID, NZ_CP051859). The read coverage was visualized using the ggplot2 R package.

**Morphological characterization of phage pBtic235 using transmission electron microscopy**. To sediment the pBtic235 particles the phage lysate (10 mL) obtained using the soft-agar-overlay method described above was centrifuged at $25,000 \times g$ for 90 min. The sediment was then washed three times with 10 mL of 0.1 M ammonium acetate buffer (pH 7), centrifuged as before, and resuspended in 300 µL of the same buffer. The phage suspension (7 µL) was deposited onto a 100-mesh copper grid (grid size, 3 mm) and covered with a carbon-coated Formvar film. After 10 min the unabsorbed liquid was removed with filter paper and 7 µL of 1% uranyl acetate was added for 10 s before the excess was removed. The copper grid containing the negatively stained phage particles was air dried at room temperature and then examined under the electron microscope (CM 100; Philips) with the images recorded by a camera (Orius; Gatan). Visualization was achieved with the support of the Infrastructural Centre for Microscopy of Biological Samples, Department of Biology, University of Ljubljana (Ljubljana, Slovenia).

**Protein expression and purification**. Constructs using the pET29b(+) expression vector were prepared by Twist Bioscience using DNA synthesis of the tectiviral gp7 homologs from phages found in *B. cereus* VD184 (GenBank ID, NZ_KB976851.1; genome coordinates, 4846 to 4995), *B. thuringiensis* serovar *aizawai* (GenBank ID, NZ_AMXT02000219.1; genome coordinates, 4557 to 4712), *B. cereus* HuB13-1 (GenBank ID, KB976635.1; genome coordinates, 4496 to 4648), *B. mycoides* VDm034 phage Lima (GenBank ID, KC152964.1; genome coordinates, 12 to 161), *B. anthracis* phage AP50 (GenBank ID, EU408779.1; genome coordinates, 4587 to 4724), *B. cereus* AND1284 phage Sato (GenBank ID, KC152965.1; genome coordinates, 13 to 150), and the *B. thuringiensis* protein M1QNS5 (GenBank ID, CP020754.1; genome coordinates, 3983220 to 3983369; UniProt ID, M1QNS5) and the *S. aureus* protein HU (GenBank ID, CP063990.1; genome coordinates, 1983142 to 1983575). Each protein was over-expressed in the *E. coli* strain BL21(DE3) (New England Biolabs, USA) as a recombinant with an N-terminal thrombin-cleavable site and a hexahistidine tag. *E. coli* strains carrying the different pET29b(+) constructs were grown aerobically at 37 °C in 500 mL of LB supplemented with 50 µg mL$^{-1}$ of kanamycin to an OD$_{600}$ of 0.6. Protein synthesis was induced by the addition of 0.8 mM IPTG. Cultures were incubated at 37 °C for 4 h, cells were pelleted and lysed, and proteins were affinity purified using Ni-chelate

chromatography (Qiagen, USA). Cell lysates were loaded onto pre-equilibrated columns (50 mM NaH$_2$PO$_4$ [pH 8.0], 300 mM NaCl, 10 mM imidazole), which were then washed (50 mM NaH$_2$PO$_4$ [pH 8.0], 300 mM NaCl, 20 imidazole), and the proteins were eluted with 50 mM NaH$_2$PO$_4$ (pH 8.0), 300 mM NaCl, 250 mM imidazole. The eluted protein fractions were dialyzed against a buffer containing 20 mM HEPES, pH 7.4, 300 mM NaCl (dialysis membrane tubing; molecular weight cut-off, 3.5 kDa; Spectrum, USA), and the protein samples were stored at −80 °C. The resulting recombinant protein concentrations were determined using a BCA Protein Assay kit (Pierce, Thermo Fisher Scientific, USA), according to the manufacturer's instructions. LexA proteins from *B. thuringiensis*, *S. aureus*, and *E. coli* and the gp7 protein of GIL01 phage were prepared and quantified as described previously[17,50,51].

**Surface plasmon resonance of protein-protein and protein-DNA interactions**. Surface plasmon resonance measurements were performed at the Infrastructural Centre for Molecular Interactions Analysis, Department of Biology, University of Ljubljana (Ljubljana, Slovenia) (T200 system; Biacore, GE Healthcare). To investigate the direct interactions of the gp7 homologs or gp7-like proteins with the LexA repressors from *B. thuringiensis*, *S. aureus*, and *E. coli*, each LexA protein was immobilized on a separate carboxymethyl-dextran-coated gold surface of a CM5 sensor chip (GE Healthcare, USA), as described previously[17]. The final amounts of immobilized ligands were 1200 to 1300 resonance units (RUs). The analytes were three-fold serially diluted (6–1458 nM) in running buffer (20 mM HEPES, pH 7.4, 150 mM NaCl, 3 mM EDTA, 0.1 mg mL$^{-1}$ bovine serum albumin, 0.005% surfactant P20), and each concentration was injected for 180 s at a flow rate of 30 µL min$^{-1}$ at 25 °C. The dissociation phase was followed for 120 s. Experiments were performed in duplicate. The sensor surface was regenerated with 0.07% sodium dodecyl sulfate for 8 s at a flow rate of 20 µL min$^{-1}$. The sensorgrams were doubly referenced for the flow cell 1 untreated surface response and the flow cell 2 buffer response. Data were analyzed using the Biacore T200 Evaluation software, and equilibrium dissociation constants (K$_D$) were determined by fitting the data to steady-state affinity models. The mean K$_D$ and standard deviations were determined from two titrations of separate experiments for each protein.

A streptavidin sensor chip (SA chip; GE Healthcare, USA) was used to study the interactions between the LexA and the gp7-like proteins and target DNA fragments, which included wild-type and mutated LexA operators within the GIL01 *P1* promoter, as well as the unique LexA operator in pBtic235 (Supplementary Table 2). To prepare the different DNA fragments, complementary primers (Sigma-Aldrich) were mixed at a mole-to-mole ratio of 1:1.5 (long:short primers) in 50 mM NaH$_2$PO$_4$, pH 7.4, 150 mM NaCl, and annealed using a temperature gradient from 95 °C to 20 °C. Approximately 40 RUs of the annealed 42-bp DNA fragments carrying a 15-nucleotide overhang were hybridized to the complementary biotinylated S1 primer (Supplementary Table 2) that was immobilized through the streptavidin–biotin interaction in flow cell 2 of the sensor chip. The DNA fragments were injected at a flow rate of 5 µL min$^{-1}$. The interactions between the LexA (5, 10, 20 nM), the small proteins (200 nM) alone, or the LexA pre-incubated with the small proteins at a molar ratio of 1:10, and the chip-immobilized DNAs were measured by injection of the protein solutions in running buffer (20 mM HEPES, pH 7.4, 150 mM NaCl, 3 mM EDTA, 0.1 mg mL$^{-1}$ bovine serum albumin, 0.005% surfactant P20) at 50 µL min$^{-1}$. The sensor surface was regenerated by injecting 50 mM NaOH at 50 µL min$^{-1}$ for 6 s.

**Bioinformatic analysis**. Alignment of the amino-acid sequences of the tectiviral gp7 homologs and the *Bacillus* spp. LexA proteins was performed using Clustal Omega and visualized using Jalview[52,53]. The tectivirus genomes were aligned and visualized using Easyfig with tblastx[54]. To detect the LexA binding sites in the pBtic235 (GenBank ID, NZ_CP051859), xFITOM[26] was used to search for sequences located up to 300 bp upstream of the translation start sites of the pBtic235 genes that shared at least 75% identity with the *Bacillus subtilis* consensus Cheo-box (5'-CGAAC(N$_4$)GTTCG-3')[25]. For the search of the gp7 structural homologs, all the available 3D protein-structure models generated by the SWISS-MODEL homology-modeling pipeline for *B. thuringiensis*, *E. coli*, and *S. aureus* were downloaded from the SWISS-MODEL Repository in April 2020[55]. Each model was aligned pair-wise with the known gp7 structure (PDB ID: 6N7O) using the ProBiS algorithm[28]. Models with >40 and <100 amino acids and Z-score of >2.0 were taken into consideration. A Z-score of 2.0 indicates a high degree of structural similarity between the gp7 model and the compared structural model, which is in the top <2% of all possible alignments of the protein structures in the Protein Data Bank[56,57]. Finally, the *B. thuringiensis* protein M1QNS5, which had the highest Z-score among all of the proteins identified, was further tested as a potential gp7 structural homolog.

**Reporting summary**. Further information on research design is available in the Nature Portfolio Reporting Summary linked to this article.

## Data availability

All of the data presented in this study are available from the corresponding author upon reasonable request. Raw RNA sequence data files have been deposited under the Bioproject with accession number PRJNA780269 and BioSample accession numbers: SAMN28961381 (GBJ002/GIL01_MMC), SAMN28961382(GBJ002/GIL01), SAMN28961383 (GBJ002_MMC), SAMN28961384(GBJ002), SAMN23134701 (GBJ002_pDG), SAMN23134702 (GBJ002_pDG7), SAMN23134703 (GBJ002_pDG_MMC), and SAMN23134704 (GBJ002_pDG7_MMC). Raw genome sequencing data files have been deposited with BioSample accession numbers: SAMN23134705 (pBtic235_WGS), SAMN30884991 (GBJ396_WGS), and SAMN30884992 (GBJ002_WGS). The Source data behind the graphs in the paper is included in Supplementary Data 7.

## Code availability

The pipeline for the RNA-seq analysis has been publicly deposited in GitHub at https://github.com/AnzeLovse/mag and the transcriptomic data analysis can be found at https://github.com/AnzeLovse/pavlin-lovse-2022.

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

## Acknowledgements

We thank Janez Kokošar for help with the pipeline for the RNA-seq analysis. This work was supported by the Slovenian Research agency (grant numbers J1-4394, J4-1778, P1-0207, and P4-0407). A.P. is a recipient of financial support from the Slovenian Research Agency for Young Researchers. Applicable funding source: Javna Agencija za Raziskovalno Dejavnost RS (Slovenian Research Agency) - J1-4394 [Butala] Javna Agencija za Raziskovalno Dejavnost RS (Slovenian Research Agency) - J4-1778 [Butala] Javna Agencija za Raziskovalno Dejavnost RS (Slovenian Research Agency) - P1-0207 [Butala] Javna Agencija za Raziskovalno Dejavnost RS (Slovenian Research Agency) – P4-0407 [Gutierrez-Aguirre].

## Author contributions

N.F., and M.B. are responsible for overall design of this study. A.P., and N.F. prepared the bacterial strains and plasmids. A.P. prepared samples for RNA-seq analysis. A.L. developed pipeline for RNA-seq analysis and analyzed the RNA-seq data. A.P., and G.B. preformed the SPR experiments. R.K., and A.P. obtained phage pBtic235 electron micrographs. J.O., and J.K. carried out the bioinformatics analysis to detect gp7-like proteins. A.K. carried out the pull-down experiments. Ž.L., I.G.A., and A.P. performed the RT-qPCR analysis and analyzed the data. M.B., A.P., A.L., and N.F. wrote the paper. All authors have given approval to the final version of the paper.

## Competing interests

The authors declare no competing interests.
