## [Peer Review File · Communications Biology]

Reviewers' comments:

Reviewer #1 (Remarks to the Author):

The manuscript by Pavlin et al. is a potentially very interesting work, focused on an enigmatic group of dual plasmid/viruses. The work may be of interest to a wide range of microbiologists but, in my opinion, several aspects should be improved.

- RNAseq method. I'm concerned about the use of only two biological replicates, whereas the standard in RNAseq is performing three independent experiments. Moreover, I find it essential that the authors provide complimentary info to validate those experiments, like the detection of gp7 protein (WB?) or the virus titer.
- The authors recurrently refer to *B. thuringiensis* serovar *isralensis*. However, in their experiments they used GBJ002, a derivative strain from 4Q7, which might be misleading. For instance, regarding the RNAseq pipeline, the authors should clarify that the reference sequences do not correspond to the GBJ002 strain genome but to the parental strain 4Q7.
- The GitHub repository does not contain details about the use of RUVs method.
- Regarding the newly obtained sequence of pBtic235 the authors should map the reads against the genome of the available assembly of HER1410 strain to remove possible contaminations before the assembly.
- Page 11 and figure 2. The authors focus on the role of gp7 related to the pBtic235 lytic cycle. However, they do not show the GIL01 induction in the absence of pBtic235. The GIL01 (and gp7) induction in a pBtic235-free strain (like HER1410) should be also shown for comparison, as it
- Bioinformatics searches for gp7 structural homologs. I have several doubts about this section. First, I wonder why they did not search for actinobacteria homologs, which might be more related than *E. coli*. Moreover, recent works suggested a common origin of firmicutes and actinobacteria phages and MGEs. I also have some concerns about the experimental approach. This section would benefit from the use of the AlphaFold database or, even better, batch generation of new structural models using alphafold or roseTTAFold pipelines. Also, structural searches would be improved using more recent methodologies, like Dali or Foldseek protocols.
- Related to the previous comment, I'd expect that as in the case of GIL01, gp7 may be encoded by MGEs. Thus, besides structural searches, the use of plasmid or MGEs protein databases using profile-profile methods would be maybe more useful. Indeed, the experimental section with gp7 homologs does not significantly contribute to the manuscript IMO.
- Page 10. The authors state that their results "indicate the participation of pBtic235 in DNA integrity maintenance and replication...". Do they mean only the host or the plasmid genome? Also, besides the proteins highlighted by the authors, there are a number of hypothetical proteins differentially expressed. Do the authors have any comments on those proteins? It would be convenient to address that, for instance trying to clarify if those proteins may be involved in some function related to the scope of the manuscript, such as defense. I'd advise the use of PADLOC or PADS databases.

Other minor points:

- Page 1. Although the authors refer to GIL01 as a replicon, in my opinion, that does not clarify one of the main characteristics of GIL01 and related betatectivirus. I'd advise further clarity that GIL01 and related viruses do not integrate into the host genome during the lysogenic cycle, rather they are maintained as a linear plasmid by their own enzymatic machinery, using a well-known protein-primed mechanism.
- Related to my previous comment, it would be interesting also to comment on the recent paper about bacteria phage-plasmid elements (Pffeifer et al. NAR 2021).
- SDS-PAGE analysis of recombinant purified proteins should be provided along with the SPR experiments.

Reviewer #2 (Remarks to the Author):

The study by Anja Pavlin and co-workers is a further exploration of both function and role of Gp7, a small LexA-interacting protein encoded by GIL01 prophage of *Bacillus thuringiensis* serovar israelensis. As was shown before (Fornelos et al 2011; Fornelos et al 2015), Gp7 can interact with the host LexA repressor and enhance its binding to operators and also, can prevent RecA-mediated auto-cleavage of LexA. In the current manuscript, the authors describe Gp7 as a global transcriptional regulator, demonstrating that its overexpression affects many SOS-related and not-related genes, phage genes and virulence genes.

That said, these results are observed under overexpression of Gp7, so it is still a question whether the native level of Gp7 affects these genes (the data presented here suggest that this is not the case, see my comments below). The authors also suggested a new function for Gp7, that is coordination of the two prophages, yet I could not find any evidence that supports this idea. In fact, the results presented in this manuscript demonstrate that the two prophages are not coordinated, as one is small, fast and jumpy (GIL01- expressing Gp7) and the other is big and slow (the jumbo phage, pBtic235).

Their findings show that GIL01 has a small genome, a high frequency of spontaneous induction and it triggers bacterial lysis after 1h. In contrast, its 'neighbour' pBtic235, has a big genome, it exhibits a much lower spontaneous induction, it is slowly induced by MMC (shown by RNA seq), and it is not lysing the cells as fast as GIL01 (Figure 2A, compare black and red lines). So, it is not clear to me, how the expression of Gp7 by GIL01 delays the induction of pBtic235? It is already delayed, raising the question how this prophage survived at all in this strain (I assume it is connected to its plasmid properties and the bacterial-related genes that it encodes for).

If the authors want to explore this hypothesis, then additional experiments are required. For example, follow pBtic235 mediated bacterial lysis in a strain that is cured of GIL01, in comparison to a strain that carry a GIL01 prophage that is deleted of its lysis genes (and maybe even some structural genes). Moreover, virion production should be monitored in these strains to see whether GIL01-Gp7 has any effect on the timing of the lytic pathway of pBtic235, and One-step growth experiments should be done. The results presented now merely demonstrate that the inhibition of the SOS response (i.e., via Gp7 interaction with LexA) results in the repression of pBtic235 induction under MMC conditions.

Of note, the data also raise the question whether Gp7 is such an efficient co-repressor of LexA, as the induction of GIL01 is high even without MMC treatment. Can the authors comment on that?

I found the RNA-seq data very confusing. Looking at Table S6, representing the transcriptome profile of str. GBJ002 (carrying both GIL01 and pBtic235) under SOS conditions, it appears that there is a negative effect of MMC on GIL01, while only 34 early genes of pBtic235 are up regulated. The data show that only two genes of GIL01 are upregulated under MMC, while the rest, including late genes, keep a steady state as that in non-MMC conditions. These findings do not reflect the lytic induction of GIL01. The authors explain that this is due to the phage high spontaneous induction. However, a 5-orders of magnitude difference is shown in virion production, in induced versus non-induced cultures. If such a difference cannot be mirrored in the RNA levels- there might be a problem with the experiment. In any case, a validation of these results using RT-qPCR analysis is required on representative genes.

Many observations described in the manuscript were done when Gp7 was episomally overexpressed from an IPTG-inducible promoter. Moreover, the *E. coli*-*Bacillus* shuttle vector (pDG148) used in this study is a multicopy replicon (Fornelos et al 2011) that could greatly support even a 'hyper-expression' of Gp7. Is it possible that an overexpressed gp7 could mediate a multiple pleiotropic effect by interacting with regulatory proteins other than LexA? Again, table S6 that presents the data of differentially expressed genes in MMC-treated str. GBJ002 carrying both prophages, GIL01 and pBtic235, does not demonstrate dramatic changes in expression of the pBtic235 genes, as was shown when Gp7 was highly overexpressed (Tables S4 and S5).

In this regard, is it possible to delete gp7? If yes, it will be nice to test the direct effect of Gp7 on pBtic235.

In general, the manuscript is not well written. In most cases the conclusions are not supported by the results, or that they are not clear (e.g., Line 311-312, Line 360-362, and Line 374-377: the conclusion here is not clear to me).

Moreover, some sections are not relevant, and can be deleted, e.g., Line 384-389, what is the contribution of this section? Line 395-409, can be summarized.

I also found some 'mysteries' regarding LexA operators described and experimentally tested in this study.

First, why the LexA operator of GIL01 is shown different than that presented in Fornelos et al 2011? The latter, 14 bp sequence 5'-CGAACAAGCGTTTT-3', is matching well (with two mismatches) the classic Cheo-box consensus (5'-CGAAC(N4)GTTCG-3'), while DNA sequence in Fig. 5C is shown as 25 bp sequence 5'-CGAACaagcGTTTTataagtGTTCG-3'. What was a reason to 'extend' the previously validated 14 bp LexA binding site by 11 additional nucleotides? As shown in Fig. 5C (the lower graph), two mutations within the Cheo-box consensus-like 5'-end of the 25-mer probe were crucial to completely eliminate interaction with LexA protein, while mutations within the extended 3'-end only partially prevented binding (Fig.5C, the upper right graph).

Second, why through the whole manuscript genomic coordinates of the unique LexA operator of pBtic235 shown as those from 196575 to 196588? Both line 368 and Fig.4 legend provide these numbers, which I think are incorrect, perhaps, a result of typos?

Fig.4C provides a Cheo-box-like sequence with one mismatch (5'-AGAACatgcGTTCG-3') that was used as a core sequence of an oligomeric probe. According to the pBtic235 genome, putative LexA binding site (5'-AGAACatgcGTTCG-3') is located from 73794 to 73807, upstream of HIS92_RS25480 (HIS92_25560). Besides Fig. 4C, there is a 'marker' of LexA binding site on the pBtic235 map (Supplementary Fig. S1) indicating its approximate position near 74 kB.

Additional comments:

line 120 should be as "B. thuringiensis serovar israelensis strain BGSC 4Q7rifR chromosome (NCBI RefSeq, NZ_CP051858.1)" etc.

lines 235-237 "Bacillus subtilis consensus LexA binding site (5'-CGAACATATGTTCG-3')", perhaps, could be replaced with "Bacillus subtilis consensus Cheo-box (5'-CGAAC(N4)GTTCG-3')", because Cheo box, as a term, is more relevant for Bacillus genetics.

line 268 Supplementary Figure S1 (a map of pBtic235) does not support a sentence regarding gene expression analysis. Perhaps, Supplementary tables should be mentioned here?

line 368 should be as "...LexA binding sequence located between coordinates 73794 and 73807 in the pBtic235 sequence..."

Fig.1 Only strain GBJ002 (that has pBtic235 alone) was used. I think strain GBJ002(GIL01) that contains both pBtic235 and GIL01 should have been used as well in this figure.

Fig.2A Lysis data of GBJ002 alone should be provided in order to evaluate the impact of the empty vector (pDG).

Fig.2B A plaque assay for GBJ002(GIL01)+MMC should be added.
Line 308: lysis is not shown by PFU. Please rephrase.

Line 311-313: To state this, dynamics of both phages should be tested with and without Gp7.
Line 330: This is a speculation. The time required for the assembly of phage particles can be determined by one-step growth infection experiments.
Line 359: pBtic genes are transcribed quite similarly to those of GIL01, as can be judged from the values of the y-axis. If the statement is valid, please think of another presentation of the data. What the authors call "early pBtic genes" seems to be scattered all over the pBtic genome, which is quite unusual for phage genomes. Please comment.
Line 375. A decrease in LexA-DNA association is evident in all concentrations tested.
Lines 385-389: The importance of these genes can be tested by generating deletion mutants and testing their ability to form plaques
Please compare the data from Fig 3, 4 and Fig 1. Are the same bacterial genes are regulated by MMC? What is the effect of lysogenization by GIL01 on gene expression of the host and of pBtic?
Verify that proteins are written as proteins, e.g. Gp7 and not gp7
Line 315: The experimental system for PFU detection should be described one paragraph earlier
Line 427-429: almost the same sentence appears twice

Reviewer #3 (Remarks to the Author):

The impact of mobile genetic elements, and particularly prophages, on their bacterial hosts is an area of increase interest to virologist and bacteriologists alike. The manuscript by Pavlin et al. presents the results of a very interesting study that uses transcriptomics, SPR and other techniques to unravel the orchestrated inductions of the lytic cycles of prophages in the entomopathogen *Bacillus thuringiensis*. This bacterium will certainly attract the attention of many biotechnologically-oriented readers. Also, the study of the cross-talk between two extrachromosomal prophages, one small linear prophage (GIL01) and one jumbo circular prophage (pBtic235), is still somewhat peculiar and adds further interest to the study.

By overexpressing in the pBtic235 lysogen the small protein gp7 from GIL01 (which was previously shown by the same laboratory that regulates GIL01 lysogenic cycle), the authors found that it down-regulates the expression of more than 250 host and pBtic235 genes and delays the induction of the latter prophage. The results suggest that GIL01 lags the induction of co-resident prophages to produce its own viral progeny after DNA damage in the host cell. In addition, the authors pinpointed a ~10 kb region that seems to be involved in regulating the lysogenic cycle of pBtic235.

In general, the article is well-written and documented, and the experimentation is well conducted. I have only minor points to be addressed by the authors:

MATERIALS & METHODS

Line 88 – The statement of the nalidixic acid use to select for *B. thuringiensis* is somewhat misleading. To my knowledge not all *B. thuringiensis* strains will be resistant to this antibiotic. Please rephrase and if necessary, adapt supplementary Table 1 as three strains of *B. thuringiensis* were used.

Line 141 – To avoid ambiguities, indicate that GIL01 lysogen is also a lysogen for pBtic235. Also indicate this in Supplementary Table 1.

Line 149 – Indicate how many biological replicates the experiment has.

Line 155 – Please rephrase the use/selection of a clear plaque mutant. "a 100 μ L clear plaque" is misleading.

RESULTS

Line 272 – move "(234 chromosomal genes, 222 pBtic235 genes)" immediately after 8.7%. Also, a

total of 5250 genes are stated in the text, but in Figures 1a-c a total of 4960 are considered for the graphs. Please explain why this difference in the text and correct if necessary.

Line 309 – it is not clear what “conditioned medium” referred to, please rephrase.

Lines 314-315 – This experiment was firstly done by the authors of reference number 39. So, this should be cited accordingly.

Line 328 – Indicate if the phage tail is contractile or not.

Lines 329-331 – The assumption that pBtic235 virion assembly is slower than GIL01 based on its morphology is somewhat speculative, as there is no data. Also, GIL01 virion possess an internal lipid membrane, that its assembly might be as complex as that of pBtic235 virion. Please remove, unless you have structural assembly data that will support this hypothesis.

Line 338 – change to “20 min after MMC treatment”

Line 382 – change to “RS25480 to RS25510 from pBtic235 were temporally”

Line 385 – remove “were”

Line 390 – Rephrase this title for something along the lines of: “Tectivirus-borne gp7 homologs enhance the DNA binding activity of LexA”

DATA AVAILABILITY

Line 470 – The Bioproject number is not correct.

FIGURES AND LEGENDS

Figure 1 – the yellow dot (for response to DNA damage) is almost not distinguishable in the figures. If possible, change the color as it is representing an important function stressed out throughout the paper.

Figure 1 legend – Indicate the number of biological replicates to maintain consistency with the other figures.

Figure 2B – It is not clear the bacterium used to assay the PFU. Was it *B. thuringiensis* HER1410? Please indicate.

Line 638 – Indicate the size of the capsid and if the tail is contractile or not.

SUPPLEMENTARY MATERIAL

Supplementary figure 1 – This figure is very close from figure 3 on reference 39 (in the main text). Please acknowledge the adaptation accordingly.

Supplementary figure 5 – The Genbank numbers of the sequences used to generate the alignments should be indicated.

Table 1 – “aureus” is not a serovar, it is a subspecies: *S. aureus* ATCC 29213 subsp. aureus.

Table 1 – HuB13-1 in plasmids and relevant features should not be italicized.

Replies to Reviewer 1 comments

Reviewer: 1

The manuscript by Pavlin et al. is a potentially very interesting work, focused on an enigmatic group of dual plasmid/viruses. The work may be of interest to a wide range of microbiologists but, in my opinion, several aspects should be improved.

- RNAseq method. I'm concerned about the use of only two biological replicates, whereas the standard in RNAseq is performing three independent experiments. Moreover, I find it essential that the authors provide complimentary info to validate those experiments, like the detection of gp7 protein (WB?) or the virus titer.

Our reply #1:

We thank the Reviewer for the comment. In the revised manuscript, we now present new RNA-seq data showing the effects of GIL01 prophage on the host transcription and pBtic235 prophage genes (new Figure 1). Experiments were performed in three biological replicates. To corroborate the RNA-seq analysis examining the effect of gp7 on the host and pBtic235 transcriptome that was performed in duplicate (data now shown in Figure 2A), we performed an additional biological replicate for each condition tested. The data presented in Figure R1 are consistent with the results shown in Figure 2A. Protein gp7 down-regulated ~230 pBtic235 genes. Because we were unable to reduce the dimensionality between the two replicates presented in the first version of the manuscript and the additional biological replicate obtained during revision using principal component analysis (PCA), we present the latter data only in this response letter (Figure R1).

Figure R1. Volcano plot showing the changes in gene expression between MMC-treated (100 ng mL⁻¹, 1 h) and IPTG-induced (0.1 mM) GBJ002(pDG7) and GBJ002(pDG) cultures (biological replicate #3). The pBtic235 genes encoded on the sense (+, plasmid module) and anti-sense (-, phage module) strands, and the host genes are shown in the indicated colors. Genes without statistically significant

fold-changes are indicated as NS. CDC, cholesterol-dependent anthrolysin O/cereolysin O family member protein and ColA, collagenase.

In addition, we have determined the GIL01 and pBtic235 phage titers (Table 1). The results are in agreement with the RNA-seq results (Figures 1 and 2A).

We have performed RT-qPCR analyses which show that gp7 mRNA levels are comparable when gp7 is expressed from the IPTG-inducible pDG7 plasmid (strain GBJ002(pDG7)) or from prophage in strain GBJ002(GIL01), both induced with MMC for 1 hour (data in Supplementary Figure S2). Of note, the expression of gp7 in the GBJ002(GIL01) strain not treated with MMC was 15-20-fold lower than in the MMC-induced strain (Supplementary Figure S2). Therefore, we omitted the data comparing the effect of ectopically expressed gp7 on the transcriptome of host GBJ002 and prophage pBtic235 in the MMC-untreated cells (former Figure 1B).

- The authors recurrently refer to *B. thuringiensis* serovar *isralensis*. However, in their experiments they used GBJ002, a derivative strain from 4Q7, which might be misleading. For instance, regarding the RNAseq pipeline, the authors should clarify that the reference sequences do not correspond to the GBJ002 strain genome but to the parental strain 4Q7.

Our reply #2:

We added this information in the Material and methods section »Pipeline for RNA-seq analysis«.

- The GitHub repository does not contain details about the use of RUVs method.

Our reply #3:

We added the missing information.

- Regarding the newly obtained sequence of pBtic235 the authors should map the reads against the genome of the available assembly of HER1410 strain to remove possible contaminations before the assembly.

Our reply #4:

As suggested by the Reviewer, we mapped the reads obtained from sequencing of the pBtic235 prophage genome to the genome of HER1410 (GenBank ID: CP050183.1) as reference. The 69.741 out of 5.249.662 reads matched the HER1410 genome. Most of these reads mapped to the HER1410 genes for tRNAs (Gln, Pro, Arg, Asp, Thr, Trp) and the ribonucleoside diphosphate reductase, *nrpE* gene. However, these are not contaminants because homologs of these genes are carried by pBtic235. Analysis shows that contamination of HER1410 sequences in the sequenced sample of pBtic235 is negligible. Therefore, we did not modify Figure 3 and Supplementary Figure S3, which show the sequencing coverage of pBtic235.

- Page 11 and figure 2. The authors focus on the role of gp7 related to the pBtic235 lytic cycle. However, they do not show the GIL01 induction in the absence of pBtic235. The GIL01 (and gp7) induction in a pBtic235-free strain (like HER1410) should be also shown for comparison, as it

Our reply #5:

Phage GIL01 is not stable in the HER1410 strain. It forms mainly lytic plaques and is therefore unable to establish a stable lysogenic state. Like other betatectiviruses that prey on *Bacillus cereus* group, GIL01 infects only certain bacterial strains from closely related hosts ¹. Furthermore, it has proven difficult to remove pBtic235 from the GBJ002 strain because of its likely functional role in bacterial chromosome replication ². Therefore, we were unable to assess the effect of GIL01 or gp7 on the behavior of the host lacking all other prophages.

- **Bioinformatics searches for gp7 structural homologs. I have several doubts about this section. First, I wonder why they did not search for actinobacteria homologs, which might be more related than *E. coli*. Moreover, recent works suggested a common origin of firmicutes and actinobacteria phages and MGEs. I also have some concerns about the experimental approach. This section would benefit from the use of the AlphaFold database or, even better, batch generation of new structural models using alphafold or roseTTAFold pipelines. Also, structural searches would be improved using more recent methodologies, like Dali or Foldseek protocols.**

- **Related to the previous comment, I'd expect that as in the case of GIL01, gp7 may be encoded by MGEs. Thus, besides structural searches, the use of plasmid or MGEs protein databases using profile-profile methods would be maybe more useful. Indeed, the experimental section with gp7 homologs does not significantly contribute to the manuscript IMO.**

Our reply #6:

We recently published that in *Acinetobacter baumannii*, the chromosomally encoded, ~7 kDa protein DdrR, a corepressor of SOS genes, directly interacts with the LexA-like protein UmuDAb ³. We show that DdrR, similar to gp7, is not a DNA binding protein. The exact molecular mechanism of DdrR coregulation remains to be elucidated. We hypothesise that gp7 proteins from phages and bacteria evolved independently (convergent evolution) and cooperate with repressors that undergo RecA-stimulated self-cleavage, such as LexA. Therefore, we did not search for gp7-like proteins in prophages related to tectiviruses, but in bacteria for which there is an indication for the existence of gp7-like proteins (the reasons for this are described in the first paragraph of the section »Tectiviruses-borne gp7 homologs function by enhancing the DNA binding activity of LexA«, i.e., in *E. coli* (MG1655 strain) and in *S. aureus* (It is of note that besides active prophages, bacteria carry a high number of degraded prophages of which determinant are expressed in cells.). Since our data show that we could not identify a gp7-like protein in these bacteria, and as also suggested by Reviewer #2, we have shortened this section and present the results of the analysis in the Supplementary Figure S4. We show the results on gp7-homologs of tectiviruses that bind and enhance LexA DNA binding activity in the main text (Figure 4).

As suggested by the Reviewer, we used AlphaFold to generate structural models. We build models for 41 of the small ORFs (40-100 residues) recently identified by the group of Joe Wade in *E. coli* ⁴. Next, we applied the Probis algorithm to identify protein models with similar structural characteristics to gp7. None of the proteins received a Z-value of at least 2 (max 1.48 for the ORF:

MKSAFRFFNSCYSHFMSALDEASSGCGYLLVEVIKTKMAAGCYPCGRFRAHIAITAA), indicating that no structural model had properties similar to gp7. The results of this analysis are not shown in the manuscript.

- **Page 10. The authors state that their results "indicate the participation of pBtic235 in DNA integrity maintenance and replication...". Do they mean only the host or the plasmid genome? Also, besides the proteins highlighted by the authors, there are a number of hypothetical proteins differentially expressed. Do the authors have any comments on those proteins? It would be**

convenient to address that, for instance trying to clarify if those proteins may be involved in some function related to the scope of the manuscript, such as defense. I'd advise the use of PADLOC or PADS databases.

Our reply #7:

We refer to the maintenance of DNA integrity of the genome and replication of plasmid pBtic235. We have added this information accordingly in the revised text (lines 106-108). The sequence of plasmid pBtic235 was last annotated in Genbank on April 14, 2022, and ORFs potentially involved in defence mechanisms were not identified. We also used the Prokaryotic Antiviral Defence Locator (PADLOC) to identify possible genes involved in viral defence on pBtic235. PADLOCK did not find any system.

Other minor points:

- **Page 1. Although the authors refer to GIL01 as a replicon, in my opinion, that does not clarify one of the main characteristics of GIL01 and related betatectivirus. I'd advise further clarity that GIL01 and related viruses do not integrate into the host genome during the lysogenic cycle, rather they are maintained as a linear plasmid by their own enzymatic machinery, using a well-known protein-primed mechanism.**

Our reply #8:

We added the information (abstract, introduction) that the prophage of GIL01 does not integrate into the chromosome during lysogeny and that it uses protein-primed replication.

- **Related to my previous comment, it would be interesting also to comment on the recent paper about bacteria phage-plasmid elements (Pffeifer et al. NAR 2021).**

Our reply #9:

We now cite the reference in the discussion section.

- **SDS-PAGE analysis of recombinant purified proteins should be provided along with the SPR experiments.**

Our reply #10:

In Supplementary Figure S6, we now show the SDS PAGE gel with all recombinant proteins used in this study.

Reviewer: 2

The study by Anja Pavlin and co-workers is a further exploration of both function and role of Gp7, a small LexA-interacting protein encoded by GIL01 prophage of *Bacillus thuringiensis* serovar israelensis. As was shown before (Fornelos et al 2011; Fornelos et al 2015), Gp7 can interact with the host LexA repressor and enhance its binding to operators and also, can prevents RecA-mediated auto-cleavage of LexA. In the current manuscript, the authors describe Gp7 as a global transcriptional regulator, demonstrating that its overexpression affects many SOS-related and not-related genes, phage genes and virulence genes.

That said, these results are observed under overexpression of Gp7, so it is still a question whether the native level of Gp7 affects these genes (the data presented here suggest that this is not the

case, see my comments below). The authors also suggested a new function for Gp7, that is coordination of the two prophages, yet I could not find any evidence that supports this idea. In fact, the results presented in this manuscript demonstrate that the two prophages are not coordinated, as one is small, fast and jumpy (GIL01- expressing Gp7) and the other is big and slow (the jumbo phage, pBtic235).

Our reply #10:

Using RT-qPCR, we show that the level of gp7 expression in the GBJ002(pDG7) strain treated with MMC and induced with IPTG for 1 hour (gp7 is expressed from the pDG7 plasmid) is comparable to gp7 expression in the MMC-induced GBJ002(GIL01) strain (gp7 expressed from the GIL01 prophage) (Supplementary Figure S2). It should be noted that qPCR analysis in the non-MMC-treated GBJ002(GIL01) strain showed that the expression of gp7 was 15-20-fold lower than in the MMC-induced strain (see also our reply #1). Therefore, we omitted the data comparing the effect of ectopically expressed gp7 on the transcriptome of host GBJ002 and prophage pBtic235 in the MMC-untreated cells (former Figure 1B), but we show the effect of gp7 expressed from the pDG7 under the conditions of SOS response (Figure 2A).

We performed additional RNA seq analyses to show that GIL01 prophage downregulates the expression of genes from pBtic235 (Figure 1B).

Their findings show that GIL01 has a small genome, a high frequency of spontaneous induction and it triggers bacterial lysis after 1h. In contrast, its 'neighbour' pBtic235, has a big genome, it exhibits a much lower spontaneous induction, it is slowly induced by MMC (shown by RNA seq), and it is not lysing the cells as fast as GIL01 (Figure 2A, compare black and red lines). So, it is not clear to me, how the expression of Gp7 by GIL01 delays the induction of pBtic235? It is already delayed, raising the question how this prophage survived at all in this strain (I assume it is connected to its plasmid properties and the bacterial-related genes that it encodes for).

Our reply #11:

In the process of reviewing this manuscript, we found that vortexing or vigorous pipetting of the solution containing phage pBtic235 inactivated the phage. Since we used such an approach to determine phage titers, this was the reason why we had not detected spontaneous induction of phage pBtic235 in the first version of the manuscript. For the revised manuscript, we mixed the phage solutions gently, which allowed us to reliably enumerate the plaque-forming units of pBtic235 (Table 1). Our results show that under the tested conditions in the exponential growth phase, the spontaneous induction of pBtic235 is approximately 7×10^4 .

If the authors want to explore this hypothesis, then additional experiments are required. For example, follow pBtic235 mediated bacterial lysis in a strain that is cured of GIL01, in comparison to a strain that carry a GIL01 prophage that is deleted of its lysis genes (and maybe even some structural genes). Moreover, virion production should be monitored in these strains to see whether GIL01-Gp7 has any effect on the timing of the lytic pathway of pBtic235, and One-step growth experiments should be done. The results presented now merely demonstrate that the inhibition of the SOS response (i.e., via Gp7 interaction with LexA) results in the repression of pBtic235 induction under MMC conditions.

Our reply #12:

We thank the Reviewer for the proposed additional experiments. We attempted to delete either the lysis genes or all the genes under the control of the P3 promoter involved in phage structure-lysis-host recognition by electroporation of DNA fragments (provided by Twist Bioscience) into the GBJ002(GIL01) host. The DNA fragments used had 500 bp homology at 5' and 3' sites to the target

site on the GIL01 genome flanking the kanamycin resistance gene. We were not successful in obtaining the GIL01 derivatives. Although the phage engineering was unsuccessful in our hands, we believe that other additional experiments performed demonstrate that under SOS response, GIL01 is the dominant prophage as it suppresses the expression of pBtic235 prophage genes.

Of note, the data also raise the question whether Gp7 is such an efficient co-repressor of LexA, as the induction of GIL01 is high even without MMC treatment. Can the authors comment on that?

Our reply #13:

We argue in the revised manuscript, that the GIL01 prophage appears to have evolved mechanisms beyond gp7 that, together with host regulatory mechanisms, form the genetic circuitry for optimal GIL01 induction under specific environmental conditions. In the revised manuscript, we describe that although the expression of gp7 from the pDG7 plasmid is close to the native level, we observe that certain different transcriptional profiles when in MMC-treated cells, gp7 is ectopically expressed or expressed from the GIL01 prophage. Thus, expression of gp7 from the pDG7 plasmid leads to the repression of certain SOS genes and activation of certain virulence factor genes, which is in contrast to the results obtained in the GBJ002(GIL01) strain. However, in the GBJ002 strain carrying either the pDG7 plasmid or the GIL01 prophage, the expression of pBtic235 genes is down-regulated. In the Discussion, we state that: "We hypothesize that certain GIL01 prophage determinants have co-evolved with gp7 to preserve the sensitivity of the cellular DNA damage response, which is important for host survival, and to fine-tune certain host functions, such as virulence-factor expression, with the lytic life cycle processes of GIL01." This is consistent with recent findings published in Cell Reports showing that: Induction of two phage elements is attenuated by the prophage repressor AriS, which interferes with the host SOS response. However, the inhibitory effect of the SOS was only observed when the AriS was ectopically overexpressed and not transcribed from prophage³².

I found the RNA-seq data very confusing. Looking at Table S6, representing the transcriptome profile of str. GBJ002 (carrying both GIL01 and pBtic235) under SOS conditions, it appears that there is a negative effect of MMC on GIL01, while only 34 early genes of pBtic235 are up regulated. The data show that only two genes of GIL01 are upregulated under MMC, while the rest, including late genes, keep a steady state as that in non-MMC conditions. These findings do not reflect the lytic induction of GIL01. The authors explain that this is due to the phage high spontaneous induction. However, a 5-orders of magnitude difference is shown in virion production, in induced versus non-induced cultures. If such a difference cannot be mirrored in the RNA levels- there might be a problem with the experiment. In any case, a validation of these results using RT-qPCR analysis is required on representative genes.

Our reply #14:

In the revised manuscript, we do not show the results of comparing the timing of gene expression in the two phages during the host SOS response (former results section "GIL01 and pBtic235 transcriptional programs after SOS induction", Figure 3, Supplementary Table S6). The reason for omitting these results was that we believe that taking culture samples at time points 0, 5, 10, 20, and 30 minutes after MMC induction likely slowed culture growth (affecting aeration and cooling the cultures slightly), which may have also affected induction of the SOS response. We performed additional transcriptomics experiments that allowed us to compare the transcriptomes of cultures that did or did not carry GIL0, both treated with MMC for 30 minutes. It is worth noting that we did not sample between 0 and 30 minutes after MMC-induction, so we did not affect the growth of the cultures with this methodological setup. The results show the up-regulation of GIL01 genes

responsible for regulation and replication, as well as structure and lysis module genes (Figure 1, Supplementary Table S2). This is consistent with the virion production results (Table 1).

Many observations described in the manuscript were done when Gp7 was episomally overexpressed from an IPTG-inducible promoter. Moreover, the E. coli-Bacillus shuttle vector (pDG148) used in this study is a multicopy replicon (Fornelos et al 2011) that could greatly support even a 'hyper-expression' of Gp7. Is it possible that an overexpressed gp7 could mediate a multiple pleiotropic effect by interacting with regulatory proteins other than LexA? Again, table S6 that presents the data of differentially expressed genes in MMC-treated str. GBJ002 carrying both prophages, GIL01 and pBtic235, does not demonstrate dramatic changes in expression of the pBtic235 genes, as was shown when Gp7 was highly overexpressed (Tables S4 and S5). In this regard, is it possible to delete gp7? If yes, it will be nice to test the direct effect of Gp7 on pBtic235.

Our reply #15:

Using qPCR, we show that one hour after addition of MMC, mRNA levels for gp7 are similar when gp7 is transcribed from the IPTG-induced pDG7 plasmid (GBJ002(pDG7)) or from GIL01 prophage (GBJ002(GIL01)). The data are shown in Supplementary Figure S2. We show the transcriptomics data collected under these conditions in the main text (Figure 2A). See also our reply #1.

Deletion of gp7 results in a lytic phenotype. Gp7 is critical for the establishment of the lysogenic life cycle. As outlined in the Results and in the Discussion sections, in addition to gp7, GIL01 phage appears to have evolved certain other determinants that regulate the induction of several SOS response and virulence factor genes. Therefore, the effect of gp7 expressed from the pDG7 or from the GIL01 prophage is different, but in both cases the expression of numerous pBtic235 genes is down-regulated.

In general, the manuscript is not well written. In most cases the conclusions are not supported by the results, or that they are not clear (e.g., Line 311-312, Line 360-362, and Line 374-377: the conclusion here is not clear to me).

Moreover, some sections are not relevant, and can be deleted, e.g., Line 384-389, what is the contribution of this section? Line 395-409, can be summarized.

Our reply #16:

We have rewritten the conclusion, lines 209-211; or we have deleted certain conclusions (former lines 360-362, were not included because of the new results presented; former lines 374-377, which previously explained that gp7 appears to dissociate from LexA when the repressor binds the operator site, were omitted).

We now summarize the results of the pull-down and *in silico* analyses performed to identify gp7-like proteins in *E. coli*, *S. aureus*, or *B. thuringiensis* (line 238). We deleted the information presented in lines 384-389 of the original manuscript.

I also found some 'mysteries' regarding LexA operators described and experimentally tested in this study.

First, why the LexA operator of GIL01 is shown different than that presented in Fornelos et al 2011? The latter, 14 bp sequence 5'-CGAACAAAGCGTTTT-3', is matching well (with two mismatches) the classic Cheo-box consensus (5'-CGAAC(N4)GTTCG-3'), while DNA sequence in Fig. 5C is shown as 25 bp sequence 5'-CGAACaagcGTTTTataagtGTTCG-3'. What was a reason to 'extend' the previously validated 14 bp LexA binding site by 11 additional nucleotides? As shown in Fig. 5C (the lower graph), two mutations within the Cheo-box consensus-like 5'-end of the 25-mer probe were crucial to completely eliminate interaction with LexA protein, while mutations within the extended 3'-end only partially prevented binding (Fig.5C, the upper right graph).

Our reply #17:

In work published in 2015 (DOI: [10.1093/nar/gkv634](https://doi.org/10.1093/nar/gkv634)), we show that LexA binds to the 14 bp long sequence 5'-CGAACaagcGTTTT-3' *dinBox1* and also to an adjacent non-canonical sequence called *dinBox1b* (5'-CGAACaagcGTTTTataagtGTTCG-3'). The binding of LexA to *dinBox1b* is unstable, and the gp7 protein stabilises the binding of LexA to *dinBox1b*. By inserting point mutations into either *dinBox1* or both *dinBox1* and *1b* sites, we tested the binding of LexA (in the presence or absence of gp7) to either both or only the *dinBox1* site. We have added the reference (DOI: [10.1093/nar/gkv634](https://doi.org/10.1093/nar/gkv634)) to the legend of Figure 4.

Second, why through the whole manuscript genomic coordinates of the unique LexA operator of pBtic235 shown as those from 196575 to 196588? Both line 368 and Fig.4 legend provide these numbers, which I think are incorrect, perhaps, a result of typos?

Fig.4C provides a Cheo-box-like sequence with one mismatch (5'-AGAACatgcGTTCG-3') that was used as a core sequence of an oligomeric probe. According to the pBtic235 genome, putative LexA binding site (5'-AGAACatgcGTTCG-3') is located from 73795 to 73807, upstream of HIS92_RS25480 (HIS92_25560). Besides Fig. 4C, there is a 'marker' of LexA binding site on the pBtic235 map (Supplementary Fig. S1) indicating its approximate position near 74 kb.

Our reply #18:

We thank the Reviewer for pointing out this error. It was our mistake, we changed the coordinates (Results section, Figure 2B legend) to 73794 – 73807.

Additional comments:

line 120 should be as "B. thuringiensis serovar israelensis strain BGSC 4Q7rifR chromosome (NCBI RefSeq, NZ_CP051858.1)" etc.

Our reply #19:

We have amended the text accordingly.

lines 235-237 "Bacillus subtilis consensus LexA binding site (5'-CGAACATATGTTCG-3')", perhaps, could be replaced with "Bacillus subtilis consensus Cheo-box (5'-CGAAC(N4)GTTCG-3')", because Cheo box, as a term, is more relevant for Bacillus genetics.

Our reply #20:

We have amended the text accordingly.

line 268 Supplementary Figure S1 (a map of pBtic235) does not support a sentence regarding gene expression analysis. Perhaps, Supplementary tables should be mentioned here?

Our reply #21:

In line 87, we now better refer the reader to the map of pBtic235 (Supplementary Figure S1).

line 368 should be as "...LexA binding sequence located between coordinates 73794 and 73807 in the pBtic235 sequence..."

Our reply #22:

Corrected.

Fig.1 Only strain GBJ002 (that has pBtic235 alone) was used. I think strain GBJ002(GIL01) that contains both pBtic235 and GIL01 should have been used as well in this figure.

Our reply #23:

In Figure 1B, we now show the analysis of transcriptomic data for the strain GBJ002(GIL01).

Fig.2A Lysis data of GBJ002 alone should be provided in order to evaluate the impact of the empty vector (pDG).

Our reply #24:

In Figure 3, we do not show the growth of GBJ002 alone because an appropriate control for strain GBJ002(pDG7) is strain GBJ002(pDG). For this reason, we did not test the possible effects of pDG on GBJ002.

Fig.2B A plaque assay for GBJ002(GIL01)+MMC should be added.

Our reply #25:

In Table 1 and in Figure 3D we now present the results obtained by plaque-assay for GBJ002(GIL01).

Line 308: lysis is not shown by PFU. Please rephrase.

Our reply #26:

We have corrected the sentence.

Line 311-313: To state this, dynamics of both phages should be tested with and without Gp7.

Our reply #27:

We modified the sentence from:

Collectively, these data show that gp7 coordinates the induction of these two bacteriophages to enable GIL01 to multiply, while delaying phage pBtic235 production under DNA damaging conditions.

To (lines 209-211):

Consistent with the observed down-regulation of pBtic235 prophage genes by the GIL01 prophage or by ectopically expressed gp7 (Figures 1B and 2A), these data show that gp7 of phage GIL01 inhibits the production of pBtic235 virions under DNA-damaging conditions.

Line 330: This is a speculation. The time required for the assembly of phage particles can be determined by one-step growth infection experiments.

Our reply #28:

We have removed the speculative sentence.

Line 359: pBtic genes are transcribed quite similarly to those of GIL01, as can be judged from the values of the y-axis. If the statement is valid, please think of another presentation of the data. What the authors call "early pBtic genes" seems to be scattered all over the pBtic genome, which is quite unusual for phage genomes. Please comment.

Our reply #29:

In the revised manuscript, we have omitted the data showing the transcription programs of GIL01 and pBtic235 after induction of the SOS response (the reasons for this decision are described in our response #14).

Line 375. A decrease in LexA-DNA association is evident in all concentrations tested.

Our reply #30:

That is correct. However, we stated in the text that LexA-DNA association signal reached similar maximum values after protein injection when LexA was injected at a saturating concentration alone or when premixed with gp7. Because a higher response (measured in RU) would be expected if gp7 remained associated with LexA when bound to DNA, this suggests that gp7 was released from the LexA-DNA complex. Because additional experiments would be required to validate this statement, we removed the statement from the manuscript.

Lines 385-389: The importance of these genes can be tested by generating deletion mutants and testing their ability to form plaques

Our reply #31:

We thank the Reviewer for the advice. We intend to perform these experiments in the near future.

Please compare the data from Fig 3, 4 and Fig 1. Are the same bacterial genes are regulated by MMC? What is the effect of lysogenization by GIL01 on gene expression of the host and of pBtic?

Our reply #32:

The data presented in former Figures 3 and 4 are not included in the revised manuscript, so we did not perform the requested analysis. We present the effects of GIL01 prophage on host and pBtic235 gene expression in Figure 1B and in Supplementary Tables S2 (MMC-treated cultures) and S3 (MMC-untreated cultures).

Verify that proteins are written as proteins, e.g. Gp7 and not gp7

Our reply #33:

In all manuscripts dealing with gp7, the gp7 protein was written as gp7 and not with the capital letter Gp7. Therefore, for consistency, we use “gp7”. Perhaps the time has come to even name the protein after its function.

Line 315: The experimental system for PFU detection should be described one paragraph earlier

Our reply #34:

We now describe this a paragraph earlier, as suggested.

Line 427-429: almost the same sentence appears twice

Our reply #35:

We have joined the two sentences into one.

Reviewer #3 (Remarks to the Author):

The impact of mobile genetic elements, and particularly prophages, on their bacterial hosts is an area of increase interest to virologist and bacteriologists alike. The manuscript by Pavlin et al. presents the results of a very interesting study that uses transcriptomics, SPR and other techniques to unravel the orchestrated inductions of the lytic cycles of prophages in the entomopathogen *Bacillus thuringiensis*. This bacterium will certainly attract the attention of many biotechnologically-oriented readers. Also, the study of the cross-talk between two extrachromosomal prophages, one small linear prophage (GIL01) and one jumbo circular prophage (pBtic235), is still somewhat peculiar and adds further interest to the study.

By overexpressing in the pBtic235 lysogen the small protein gp7 from GIL01 (which was previously shown by the same laboratory that regulates GIL01 lysogenic cycle), the authors found that it down-regulates the expression of more than 250 host and pBtic235 genes and delays the induction of the latter prophage. The results suggest that GIL01 lags the induction of co-resident prophages to produce its own viral progeny after DNA damage in the host cell. In addition, the authors pinpointed a ~10 kb region that seems to be involved in regulating the lysogenic cycle of pBtic235.

In general, the article is well-written and documented, and the experimentation is well conducted. I have only minor points to be addressed by the authors:

Our reply #36:

We thank the Reviewer for positive and constructive comments.

MATERIALS & METHODS

Line 88 – The statement of the nalidixic acid use to select for *B. thuringiensis* is somewhat misleading. To my knowledge not all *B. thuringiensis* strains will be resistant to this antibiotic. Please rephrase and if necessary, adapt supplementary Table 1 as three strains of *B. thuringiensis* were used.

Our reply #37:

With the exception of HER1410, all strains of *Bacillus thuringiensis* used in this study carry chromosomal resistance to nalidixic acid, as they are all derivatives of GBJ002 with this property. Nalidixic acid was used only in plaque assays that we repeated and expanded for revision purposes. In this case, we omitted the use of nalidixic acid to make the experiments comparable to the transcriptome analysis setup. Therefore, nalidixic acid was not used in the latest setup, and we have updated the relevant Materials and Methods sections in the revised manuscript.

Line 141 – To avoid ambiguities, indicate that GIL01 lysogen is also a lysogen for pBtic235. Also indicate this in Supplementary Table 1.

Our reply #38:

We thank the Reviewer for this comment and have added the information to the text and Supplementary table S7 (former Supplementary Table S1) accordingly.

Line 149 – Indicate how many biological replicates the experiment has.

Our reply #39:

We have inserted the missing information in the revised Material and Methods section.

Line 155 – Please rephrase the use/selection of a clear plaque mutant. “a 100 μ L clear plaque” is misleading.

Our reply #40:

We have rephrased the sentence (line 386).

RESULTS

Line 272 – move “(234 chromosomal genes, 222 pBtic235 genes)” immediately after 8.7%. Also, a total of 5250 genes are stated in the text, but in Figures 1a-c a total of 4960 are considered for the graphs. Please explain why this difference in the text and correct if necessary.

Our reply #41:

We have moved the text in the brackets as recommended. Before differential expression analysis, non-expressed genes were filtered out by requiring more than five reads per gene in at least two samples. This accounts for the difference in the total number of genes (of the genome). We explain

this in the description of the RNA-seq analysis. We now also present these data (actual number of total genes) in the Figures 1A, 1B and 2A.

Line 309 – it is not clear what “conditioned medium” referred to, please rephrase.

Our reply #42:

We have rephrased the sentence to clarify the matter.

Lines 314-315 – This experiment was firstly done by the authors of reference number 39. So, this should be cited accordingly.

Our reply #43:

We have inserted the missing reference (line 201).

Line 328 – Indicate if the phage tail is contractile or not.

Our reply #44:

We have not indicated in the text whether the tail of phage pBtic235 is contractile or not, because we have not studied this question in detail.

Lines 329-331 – The assumption that pBtic235 virion assembly is slower than GIL01 based on its morphology is somewhat speculative, as there is no data. Also, GIL01 virion possess an internal lipid membrane, that its assembly might be as complex as that of pBtic235 virion. Please remove, unless you have structural assembly data that will support this hypothesis.

Our reply #45:

We have omitted this statement in the revised version of the manuscript.

Line 338 – change to “20 min after MMC treatment”

Our reply #46:

This sentence has been removed in the revised version of the manuscript because we have replaced the results showing the transcriptome of GBJ002(GIL01) 20 minutes after addition of MMC to bacteria in the experimental growth phase with the results of experiments performed in three biological replicates showing transcriptomes in GBJ002(GIL01) 30 minutes after addition of MMC (new Figure 1B).

Line 382 – change to “RS25480 to RS25510 from pBtic235 were temporally”

Line 385 – remove “were”

Our reply #47:

We have changed the text accordingly.

Line 390 – Rephrase this title for something along the lines of: “Tectivirus-borne gp7 homologs enhance the DNA binding activity of LexA”

Our reply #48:

We have rephrased the title as suggested.

DATA AVAILABILITY

Line 470 – The Bioproject number is not correct.

Our reply #49:

We have corrected the Bioproject number to PRJNA780269 (this number will become active when the manuscript is accepted). The data can be accessed *via* the link in the “Data Availability” section: <https://dataview.ncbi.nlm.nih.gov/object/PRJNA780269?reviewer=8t7iaetkh5vr1mfbpktbalfqrb>.

FIGURES AND LEGENDS

Figure 1 – the yellow dot (for response to DNA damage) is almost not distinguishable in the figures. If possible, change the color as it is representing an important function stressed out throughout the paper.

Our reply #50:

We omitted the color distinction for genes belonging to the DNA damage response because several of the yellow-labeled genes were pBtic235 homologs of host SOS genes (such as RecA, LigA). We therefore used a simplified color coding: light and dark blue for pBtic235 and gray for the host.

Figure 1 legend – Indicate the number of biological replicates to maintain consistency with the other figures.

Our reply #51:

We have inserted the missing information.

Figure 2B – It is not clear the bacterium used to assay the PFU. Was it *B. thuringiensis* HER1410? Please indicate.

Our reply #52:

We have inserted the missing information.

Line 638 – Indicate the size of the capsid and if the tail is contractile or not.

Our reply #53:

We have inserted the missing information regarding the size of the capsid. We do not have sufficient data to state whether the pBtic235 phage carries a contractile tail or not.

SUPPLEMENTARY MATERIAL

Supplementary figure 1 – This figure is very close from figure 3 on reference 39 (in the main text). Please acknowledge the adaptation accordingly.

Our reply #54:

We have added the missing reference.

Supplementary figure 5 – The Genbank numbers of the sequences used to generate the alignments should be indicated.

Our reply #55:

We have added the missing information.

Table 1 – “aureus” is not a serovar, it is a subspecies: *S. aureus* ATCC 29213 subsp. aureus.

Table 1 – HuB13-1 in plasmids and relevant features should not be italicized.

Our reply #56:

We thank the reviewer for these observations. We have corrected the table accordingly.

Reviewers' comments:

Reviewer #1 (Remarks to the Author):

I appreciate the effort made by the authors in the revised version. Overall, they have done a great job and answered most of my concerns.

I only have some minor doubts regarding reply #6. First, they did not answer why actinobacteria homologs were not searched. I didn't mean to search in actinobacteria phages, but in actinobacteria proteomes.

Second, the use of ProBiS method does not provide anything new. As I understood, they only compare the small ORFs from ref. 4. Finding homologies between *E. coli*/*Acinetobacter* and *Bacillus* proteins is clearly very challenging and it may be a risky bet. Why they didn't check a whole protein database? Did they use the alphafold database?

Reviewer #2 (Remarks to the Author):

The revised manuscript, is in fact a new manuscript.

The only conclusion the authors can surely state is that "gp7 represses the induction of pBtic235".

1. Abstract line31: replace "delay" with "repress", as no temporal resolution is shown in the manuscript.
2. line 31-32 " to allow GIL01 to selectively produce its own progeny" - this is only a speculation, there is no data to support this statement. Anyway, GIL01 lyses the cells in 1 h, while pBtic starts to do so only 2 hours later. In this regard, it is of interest to test whether a timely expression of gp7 e.g., for one hour (and then removing the inducer) will have a similar effect on pBtic production as all-course ectopic expression of gp7.
3. The proposed mechanism of gp7 effect on pBtic is via the regulation of LexA. However, more effort should be done to connect LexA to pBtic. For example, one can first show that pBtic induction indeed depends on LexA de-repression, and then combine LexA3 mutation with gp7 overexpression, to see whether indeed gp7 effect is mediated by LexA and not by its pleotropic function.
4. Fig. 2C. and line 194: gp7 represses not only the operon downstream to the proposed LexA binding site (as expected), but also two other clusters in the region presented: genes 4-9 (upstream to LexA box), 17-19 and 20. Please comment.
5. line 205: "pBtic235-induced lysis" - data supporting this statement should be shown explicitly in the figure or appropriate reference should be provided.
6. I do not understand how the paragraph about the clear plaques is related to the topic of the manuscript. It is a stand-alone section that does not contribute to the understanding of the phenomenon. In any case, more information should be provided about these 12 genes: please provide the list and the annotations as a separate table, single gene deletions may be required to find the gene responsible for clear plaques and more..

Reviewer #1:

I appreciate the effort made by the authors in the revised version. Overall, they have done a great job and answered most of my concerns.

I only have some minor doubts regarding reply #6. First, they did not answer why actinobacteria homologs were not searched. I didn't mean to search in actinobacteria phages, but in actinobacteria proteomes.

Second, the use of ProBiS method does not provide anything new. As I understood, they only compare the small ORFs from ref. 4. Finding homologies between *E. coli*/Acinetobacter and Bacillus proteins is clearly very challenging and it may be a risky bet. Why they didn't check a whole protein database? Did they use the alphafold database?

Our reply #1: We thank the reviewer for the positive comments. As stated in our response to revision #1 and in the manuscript (lines 233-236: "In a previous study we established that gp7 can interact with the LexA of *S. aureus* and enhance its DNA binding¹⁷. Therefore, proteins with similar functional properties to gp7 might modulate the LexA activity in *S. aureus*. This might also be the case in *E. coli*, where an unidentified factor appears to be required for LexA binding to multiple target sites²⁷."), we searched for the gp7 homologs in bacteria for which there was an indication for the existence of gp7-like proteins. Therefore, we did not check the PDB structures and models of protein structures of actinobacteria. As advised by the reviewer, we also used ProBiS to search the entire alphafold *E. coli* and *S. aureus* database for proteins comprising 40-100 amino acid residues with surface structural properties similar to gp7. Protein structures with a Z-score >2.2 were considered.

The search found gp7-like proteins in *S. aureus* (Uniprot IDs): Q2FV68 (Z score of 2.34; 55 amino acids), Q2FW55 (Z score of 2.33; 55 amino acids), Q2FV46 (Z score of 2.22; 76 amino acids), and Q2FY18 (Z score of 2.20; 50 amino acids). In addition, the analysis also highlighted protein PODPM9 (Z-score of 2.56; 74 amino acids) from *E. coli*.

We ordered (Twist Biosciences) expression plasmids carrying a T7 promoter that controls the expression of each of the above genes. We used the *E. coli* BL21(DE3) expression strain and applied the conditions described in the manuscript for purification of the tectiviral gp7 homologs. We succeeded in isolating only the *E. coli* (PODPM9) protein (Figure R1). Next, using SPR and conditions described in the manuscript, we tested whether the PODPM9 protein enhances DNA binding of *E. coli* LexA to its cognate *recA* operator, but the protein had no measurable effect on LexA (Figure R2). We present these data only in this response letter.

Figure R1: SDS-PAGE analysis of the dialyzed elution fractions of proteins Q2FV68, Q2FW55, Q2FV46, Q2FYI8 or PODPM9. Each protein fraction and PageRuler™ prestained protein ladder (Thermo Scientific™) were resolved on SurePAGE™ 4 to 20% bis-Tris protein gels (GenScript) and stained with SimplyBlue SafeStain (Invitrogen).

Figure R2: Surface plasmon resonance sensorgrams of the interaction between *E. coli* LexA alone (red line) or pre-incubated with protein PODPM9 with the chip-immobilized *recA* operator. Free LexA (10 nM) or LexA (10 nM) pre-incubated with PODPM9 protein in 5, 10 or 100 nM concentrations were injected over the 30 RU of immobilized DNA at 50 $\mu\text{L min}^{-1}$. The experiment was performed in duplicate and representative sensorgrams are shown.

Reviewer #2:

The revised manuscript, is in fact a new manuscript.

The only conclusion the authors can surely state is that " gp7 represses the induction of pBtic235".

1. Abstract line31: replace "delay" with "repress", as no temporal resolution is shown in the manuscript.

Our reply #2: We thank the reviewer for this comment. We modified the manuscript to state that gp7 represses the induction of phage pBtic235.

2. line 31-32 " to allow GIL01 to selectively produce its own progeny" - this is only a speculation, there is no data to support this statement. Anyway, GIL01 lyses the cells in 1 h, while pBtic starts to do so only 2 hours later. In this regard, it is of interest to test whether a timely expression of gp7 e.g., for one hour (and then removing the inducer) will have a similar effect on pBtic production as all-course ectopic expression of gp7.

Our reply #3: We are grateful to the reviewer for their insightful suggestion, as we considered carrying out this experiment too. To closely monitor the expression of gp7 in the host, in absence of and throughout induction, we would ideally need anti-gp7 antibodies. Our attempts to immunize mice against gp7 [purified gp7 in PBS buffer was injected into three mice at a concentration of 0.5 mg/mL] were unsuccessful since the initial and booster gp7 shots did not elicit a sufficiently strong immune response. The mice sera tested negative for gp7 in dot-blot and Western blot analyses. Furthermore, the autopsy of the immunized mice revealed enlarged spleens and livers. We do not feel confident reporting on the pBtic induction timing without measuring gp7 levels in the cell. This is because we do not know the lifetime of gp7 in the cell as well as how long it persists inside the cell after the inducer is removed. We therefore propose adopting a more conservative approach by modifying our original statement as follows: “to **presumably** allow GIL01 to **multiply first**”.

3. The proposed mechanism of gp7 effect on pBtic is via the regulation of LexA. However, more effort should be done to connect LexA to pBtic. For example, one can first show that pBtic induction indeed depends on LexA de-repression, and then combine LexA3 mutation with gp7 overexpression, to see whether indeed gp7 effect is mediated by LexA and not by its pleotropic function.

Our reply #4:

To further describe the role of LexA at pBtic235, we used a GBJ002 strain carrying a noncleavable *lexA(A96D)* mutation (strain GBJ396¹). We sequenced strains GBJ002 and GBJ396 (SRA IDs: SAMN30884991 and SAMN30884992). We used Snippy (v3.2)² for alignment and variant calling from the paired-end whole-genome sequencing reads of GBJ002 and GBJ396 strains. For processing, we used the reference sequence of BGSC 4Q7rifR (NCBI RefSeq, NZ_CP051858.1) and the default parameters. A snapshot of the Conda environment used for the analysis is deposited on GitHub <https://github.com/AnzeLovse/pavlin-lovse-2022>. Alignment of the sequencing reads with the genome of *B. thuringiensis* serovar *israelensis* strain BGSC 4Q7rifR (NCBI RefSeq, NZ_CP051858.1) showed that GBJ396, in contrast to GBJ002, in addition to the nonsynonymous mutation in *lexA (A96D)* that renders LexA incapable of self-cleavage, also carries an approximately 5 kbp deletion (missing genes with annotations according to the strain BGSC 4Q7rifR): RS00265 (spore coat protein), RS00270 (spore coat protein), RS00275 (spore coat protein), RS00280 (GNAT family N-acetyltransferase), RS00285 (no annotated function), RS00290 (no annotated function), and RS00295 (no annotated function). In addition, the GBJ396 strain carries a nonsense mutation in the gene for arginine decarboxylase (G89S) and a frameshift mutation in the gene for farnesyl diphosphate synthase. These mutations could have arisen during construction of the mutant strain to compensate for a fitness deficiency inflicted by the *lexA(A96D)* mutation.

Importantly, we also found that the GBJ396 strain lacks the pBtic235 prophage since no sequencing reads mapped the pBtic235 plasmid (GenBank ID: NZ_CP051859). Considering the absence of pBtic235 in GBJ396, we attempted to infect the GBJ396 strain with pBtic235 but were unsuccessful. To do this we spotted a suspension of phage pBtic235 at a concentration of 7×10^4 PFU/mL onto the lawns of the *B. thuringiensis* HER1410 or GBJ396. While turbid plaques were observed on HER1410, no plaques were observed on the lawn of strain GBJ396. We used PCR and pBtic235-specific primers³ and confirmed that pBtic235 plasmid was not present in GBJ396, making the study of pBtic235 prophage induction in the LexA3 strain not conceivable. We added this information in lines 214-219 (page 8).

Nevertheless, curiosity drove us to carry out growth and induction experiments in strains GBJ002 and GBJ396. When we added 100 ng mL⁻¹ MMC to strain GBJ396, the strain did not lyse (Figure R3). This strongly suggests that lysis in strain GBJ002 after the addition of MMC is a consequence of induction of pBtic235 prophage.

Figure R3: Comparison of growth kinetics of *B. thuringiensis* strains GBJ002 (black), GBJ002 *lexA(A96D)* (red), and GBJ002 *lexA(A96D)* (pDG7) (green). IPTG (0.1 mM) was added to GBJ002 *lexA(A96D)* (pDG7) cultures ~2 h before time zero. MMC (100 ng mL⁻¹) was added to half of the cultures in the early exponential phase (OD₅₉₅ ~0.4) at time zero. Bacterial density of the MMC-treated cultures is shown with solid lines, and that of the untreated cultures is shown with dotted lines. Bacterial growth was measured for 3 h in 1-h intervals.

4. Fig. 2C. and line 194: gp7 represses not only the operon downstream to the proposed LexA binding site (as expected), but also two other clusters in the region presented: genes 4-9 (upstream to LexA box), 17-19 and 20. Please comment.

Our reply #5:

It is conceivable that among the pBtic235 genes controlled by LexA/gp7, there are also regulators that affect the induction of several pBtic235 genes. For example, LexA/gp7 could repress an activator that controls the expression of pBtic235 genes that are thus indirectly controlled by LexA/gp7. We believe that pBtic235 has evolved a regulatory cascade in which one of the first regulatory levels is repressed by LexA and gp7.

To suggest this, we modified the paragraphs on page 7, lines 189-198:

“The identified LexA target site was located upstream of a putative operon that consists of seven ORFs (gene IDs, RS25480 to RS25510), one of which showed homology with the nucleoside 2-deoxyribosyltransferase (RS25495) and another with metallophosphoesterase (RS25505). The sequences of the remaining ORFs did not match any genes with known functions. A closer look at the

transcriptomic data for GBJ002(pDG7) (Figure 2A, Supplementary Table S5) revealed that ORFs RS25480 to RS25510 were repressed by gp7 (Figure 2C). Together, the SPR spectrometry and transcriptomic data suggested that these ORFs are organized into an operon that is repressed by LexA and its co-repressor gp7.

In presence of gp7, we also observed the repression of genes RS25450 to RS25475 and RS25515 to RS25530 (Figure 2C) despite the absence of a detectable LexA binding site upstream these genes or their presumed operons. It is therefore possible that ORFs RS25480 to RS25510 encode a regulator controlling the expression of those genes or that gp7 enables LexA binding to poorly conserved sites, as observed in phage GIL01.”

5. line 205: "pBtic235-induced lysis" - data supporting this statement should be shown explicitly in the figure or appropriate reference should be provided.

Our reply #6: Growth of strain GBJ396, which lacks pBtic235, is not affected by the addition of a subinhibitory concentration of MMC (Figure R3, please refer to our reply #4), compared with strain GBJ002 in the presence of MMC (Figure 3A). This suggests that induction of pBtic235 prophage causes cell lysis. These data are also consistent with the results in Figure 3A-3C, which show that the enhanced decrease in bacterial density after MMC-treated culture GBJ002 is proportional to the higher pBtic235 phage titers.

6. I do not understand how the paragraph about the clear plaques is related to the topic of the manuscript. It is a stand-alone section that does not contribute to the understanding of the phenomenon. In any case, more information should be provided about these 12 genes: please provide the list and the annotations as a separate table, single gene deletions may be required to find the gene responsible for clear plaques and more.

Our reply #7:

We partially agree with the reviewer that the pBtic235 clear-plaque mutant result is somewhat detached from the main thread, but we believe that this result is important to the description of the pBtic235 cycle and for future research on pBtic235 and related prophages. Therefore, we moved the results of the clear-plaque pBtic235 mutant (formerly Figure 3E) to the supplementary material (modified Supplementary Figure S3) and changed the manuscript accordingly. In Figure S3, we also added a table with information on 12 genes deleted in the genome of the clear-plaque mutant.

that of the untreated cultures is shown with dotted lines. Bacterial growth was measured for 5 h in 1-h intervals or until complete cell lysis. Black arrows indicate the times at which the culture samples were collected for determination of phage counts in (B), (C) and (D). Each data point represents the mean of three replicates. Error bars indicate standard deviations. (B, C, D) Spot titers of MMC-treated culture supernatants of strains GBJ002(pDG7) (B), GBJ002(pDG) (C) and GBJ002(GIL01) (D) on the recipient host, *B. thuringiensis* HER1410. The experiment was performed in duplicate. (E) Representative transmission electron micrograph of the pBtic235 phage showing an isometric head of ~80-nm diameter and an ~200-nm-long tail. Scale bar, 100 nm.

Former Supplementary Figure S3. Sequencing depth of the pBtic235 clear-plaque variant. pBtic235 sequencing depth (read coverage in gray). pBtic235 genes encoded on sense and anti-sense genes are given in red and blue, respectively.

B

Gene ID	Protein ID	Aminoacid sequence length	NCBI homology based annotation	InterPro protein domain search
HIS92_RS26165	WP_000424042.1	507	tRNA nucleotidyltransferase	NA
HIS92_RS26170	WP_001071098.1	242	Hypothetical phage protein	None predicted
HIS92_26225	WP_000589206.1	51	Hypothetical protein	None predicted
HIS92_RS26175	WP_000502605.1	97	Hypothetical protein	None predicted
HIS92_RS26180	WP_000108739.1	524	Hypothetical protein	None predicted
HIS92_RS26185	WP_000149726.1	72	Hypothetical protein	None predicted
HIS92_RS26190	WP_000229935.1	241	Hypothetical protein	None predicted
HIS92_RS26195	WP_000455658.1	99	SpoVG family protein	NA
HIS92_RS26200	WP_000783539.1	155	Hypothetical protein	None predicted
HIS92_RS26205	WP_001097001.1	174	Hypothetical protein	None predicted
HIS92_RS26210	WP_000174333.1	313	Hypothetical protein	Actin-like ATPase domain
HIS92_RS26215	WP_000039126.1	383	Replication initiator protein A	NA

*NA (Not Assessed)

New Supplementary Figure S3. The deleted region of the pBtic235 clear-plaque variant. (A) Sequencing depth of the deleted genomic region and its flanking regions (number of reads matching to the region from 212 to 230 kb) in the clear-plaque pBtic235 variant. Gene ID numbers correspond to the new locus tags, except for the old locus tag for gene “26255”, which is not annotated in the reference genome sequence (GenBank ID, NZ_CP051859). The pBtic235 sequencing depth (read coverage) is shown in gray.

pBtic235 genes encoded on sense and anti-sense strands are given in red and blue, respectively. **(B)** List of deleted genes in the clear-plaque pBtic235 variant.

References:

1. Fornelos, N., Bamford, J. K. H. & Mahillon, J. Phage-borne factors and host LexA regulate the lytic switch in phage GIL01. *J. Bacteriol.* **193**, 6008–19 (2011).
2. Seemann, Torsten. Snippy: fast bacterial variant calling from NGS reads. <https://github.com/tseemann/snippy>, (2015).
3. Gillis, A. *et al.* Detection of the cryptic prophage-like molecule pBtic235 in *Bacillus thuringiensis* subsp. *israelensis*. *Res. Microbiol.* **168**, 319–330 (2017).

REVIEWERS' COMMENTS:

Reviewer #1 (Remarks to the Author):

The authors have answered all my previous concerns. I have no further comments.

Reviewer #2 (Remarks to the Author):

My comments were addressed. I accept.